# A unique symbiosome in an anaerobic single-celled eukaryote

Jon Jerlström-Hultqvist [1,2] ✉, Lucie Gallot-Lavallée [2],
Dayana E. Salas-Leiva [2,3], Bruce A. Curtis[2], Kristína Záhonová [4,5,6,7],
Ivan Čepička [8], Courtney W. Stairs [9], Shweta Pipaliya[4], Joel B. Dacks [4,5,10],
John M. Archibald [2] & Andrew J. Roger [2] ✉

Symbiotic relationships between eukaryotes and prokaryotes played pivotal roles in the evolution of life and drove the emergence of specialized symbiotic structures in animals, plants and fungi. The host-evolved symbiotic structures of microbial eukaryotes – the vast majority of such hosts in nature – remain largely unstudied. Here we describe highly structured symbiosomes within three free-living anaerobic protists (*Anaeramoeba* spp.). We dissect this symbiosis using complete genome sequencing and transcriptomics of host and symbiont cells coupled with fluorescence in situ hybridization, and 3D reconstruction using focused-ion-beam scanning electron microscopy. The emergence of the symbiosome is underpinned by expansion of gene families encoding regulators of membrane trafficking and phagosomal maturation and extensive bacteria-to-eukaryote lateral transfer. The symbionts reside deep within a symbiosomal membrane network that enables metabolic syntrophy by precisely positioning sulfate-reducing bacteria alongside host hydrogenosomes. Importantly, the symbionts maintain connections to the *Anaeramoeba* plasma membrane, blurring traditional boundaries between ecto- and endosymbiosis.

Symbioses between eukaryotes and prokaryotes are important sources of novelty and drivers of evolutionary change. In a variety of multicellular eukaryote lineages, host body plans and physiologies have been adapted to support such interactions including the evolution of specialized symbiotic "organs"[1]. If the symbionts are intracellular, such structures are often referred to as "symbiosomes" that have been defined as "membrane-bound compartment[s] housing one or more symbionts ... located in the cytoplasm of eukaryotic cells"[2].

Symbiosomes have been described in a variety of symbiotic contexts such as in the root-nodules of plants[1], aphid bacteriocytes[1] and in protists[2]. Although many diverse unicellular eukaryote (protist) lineages are also known to host prokaryotic partners, the host-cell adaptations to support such symbiotic interactions are generally poorly understood.

In oxygen-poor environments, protists are sometimes found in syntrophic partnerships with ecto- or endosymbiotic prokaryotes[3].

[1]Department of Cell and Molecular Biology, Uppsala Universitet, Uppsala, Sweden. [2]Institute for Comparative Genomics, Department of Biochemistry and Molecular Biology, Dalhousie University, Halifax, NS, Canada. [3]Department of Biochemistry, University of Cambridge, Cambridge, UK. [4]Division of Infectious Diseases, Department of Medicine, Faculty of Medicine and Dentistry, and Department of Biological Sciences, University of Alberta, Edmonton, AB, Canada. [5]Institute of Parasitology, Biology Centre, Czech Academy of Sciences, České Budějovice (Budweis), Czechia. [6]Department of Parasitology, Faculty of Science, Charles University, BIOCEV, Vestec, Czechia. [7]Life Science Research Centre, Department of Biology and Ecology, Faculty of Science, University of Ostrava, Ostrava, Czechia. [8]Department of Zoology, Faculty of Science, Charles University, Prague, Czechia. [9]Department of Biology, Lund University, Lund, Sweden. [10]Centre for Life's Origin and Evolution, Department of Genetics, Evolution, & Environment, University College, London, UK. ✉e-mail: jon.jerlstrom.hultqvist@icm.uu.se; andrew.roger@dal.ca

These interactions are usually centered on the exchange of metabolites produced by the host's mitochondrion-related organelles (MROs), that perform metabolism adapted to anaerobic conditions. $H_2$ produced by these "hydrogenosome"-type MROs is an important currency in these interactions and is the end-product of an anaerobic ATP-producing metabolic pathway. Removal of $H_2$ by prokaryotes likely increases the metabolic flux of the host by avoiding product inhibition of anaerobic metabolism. As a result, $H_2$-consuming symbionts are sometimes found associated with host MROs, an arrangement that increases their access to substrate[4]. Although symbiont-associated subcellular structures have occasionally been observed in such anaerobic protists (e.g., ref. [5]), little is known about these systems other than the metabolites likely being exchanged between host and symbionts.

In this study, we comprehensively investigate a highly organized symbiosis involving Anaeramoebae, a recently described phylum of anaerobic protists within the Metamonada supergroup[6,7]. Anaeramoebid cells are predominantly ameboid, containing a densely packed mass of symbionts intricately interwoven with double-membraned electron-dense MROs we have recently identified as hydrogenosomes[6]. These symbionts, although not in direct contact with the hydrogenosomes, are enveloped by membranes of host origin (Fig. 1)[7].

By employing a combination of comparative genomic and transcriptomic analyses for both host and symbionts, phylogenomic investigations, focused-ion-beam scanning electron microscopy (FIB-SEM) tomography, and in situ localization techniques, we systematically dissect the intricate metabolic interactions between Anaeramoebae and their symbionts. Furthermore, we delve into the structural aspects of the host-cell-derived symbiosome and explore the evolutionary roots of this distinctive symbiotic framework. Specifically, we: (i) reconstruct the symbiosome membrane network, (ii) identify the bacterial symbionts (iii) characterize the complete genomes of three host Anaeramoebae and each of their associated symbionts, (iv) reconstruct the host-symbiont metabolic and cellular interactions, and (v) delineate the evolutionary trajectories of host and symbiont genomes.

Our findings reveal that the symbionts are situated within a symbiosome, which comprises an interconnected host membrane network that positions the symbionts alongside host hydrogenosomes. This network also features tubular connections that connect to the plasma membrane allowing access to the extracellular environment. Notably, lateral gene transfer (LGT) and extensive gene duplication emerge as pivotal factors that established and stabilized the incipient symbiosome within the ancestral lineage of Anaeramoebae.

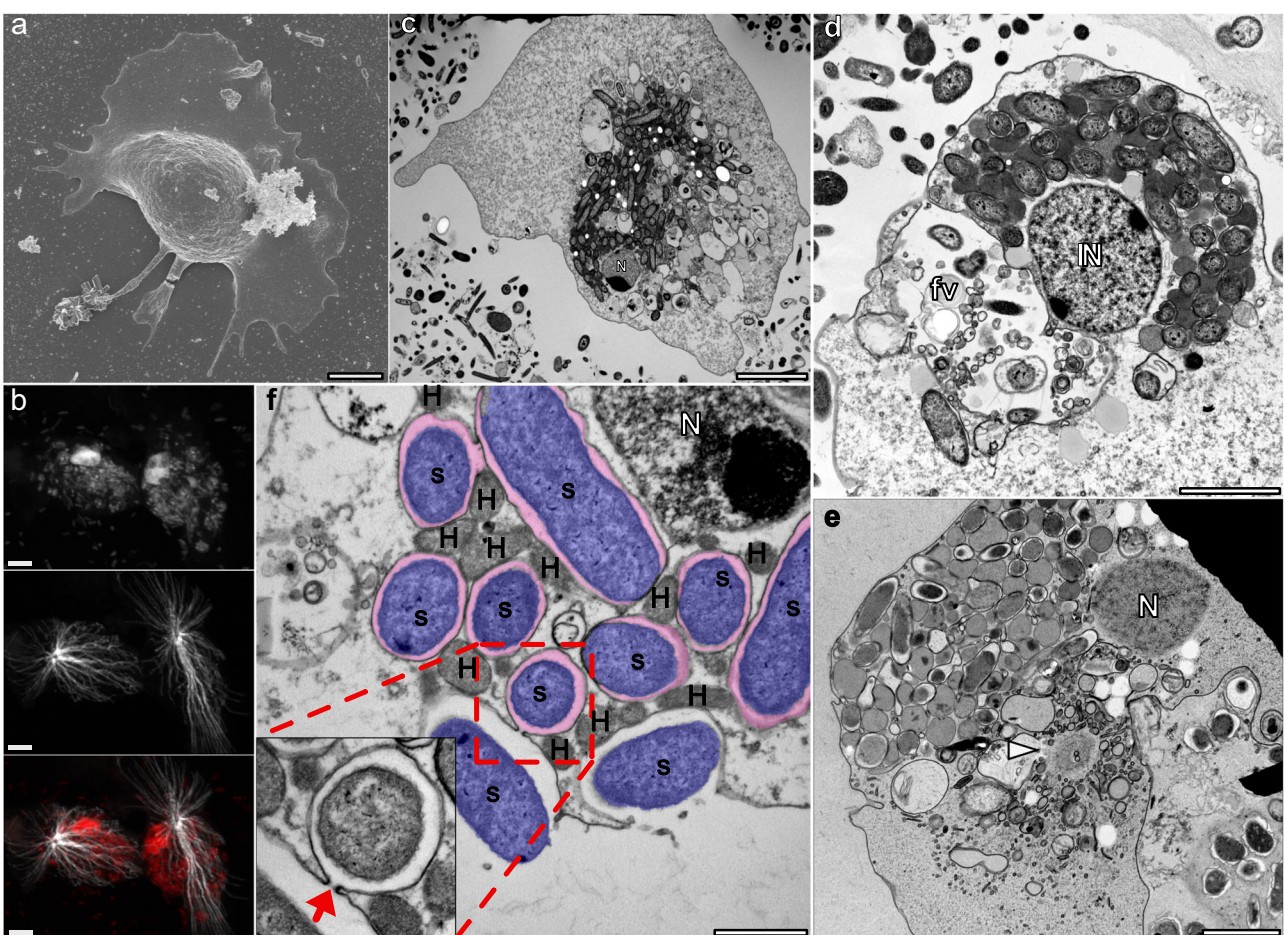

**Fig. 1 | *Anaeramoeba* and symbionts. a** Scanning electron micrograph (SEM) of an *Anaeramoeba flamelloides* BUSSELTON2 amoebae showing a hyaline front and posterior trailing projections. Scale bar 5 μm. **b** Immunolocalization of alpha-tubulin (TAT1 antibody, 1:200) in *A. flamelloides* BUSSELTON2 cells using laser scanning confocal microscopy. Host nuclei and symbiont DNA were stained using DAPI (top panel) and the acentriolar centrosome and radiating microtubules by alpha-tubulin TAT1 antibody (middle panel) with overlay (bottom panel). Scale bar 5 μm. **c** TEM of chemically fixed *A. flamelloides* BUSSELTON2. Scale bar 5 μm. **d** TEM of chemically fixed *A. flamelloides* BUSSELTON2. Scale bar 2 μm. **e** TEM of cryo-fixed *A. flamelloides* BUSSELTON2. White arrow shows acentriolar centrosome. Scale bar 2 μm. **f** Transmission electron micrograph (TEM) of *A. ignava* BMAN showing vesicle-bound symbionts (S) and host hydrogenosomes (H) in close proximity. The boxed inlay shows narrow openings (red arrow) connecting outside media to the vesicles housing the symbionts. Scale bar 1 μm. Nucleus (N), Food-vacuole (fv). The experiment in (**b**) shows representative cells from three independent replicate immunostainings.

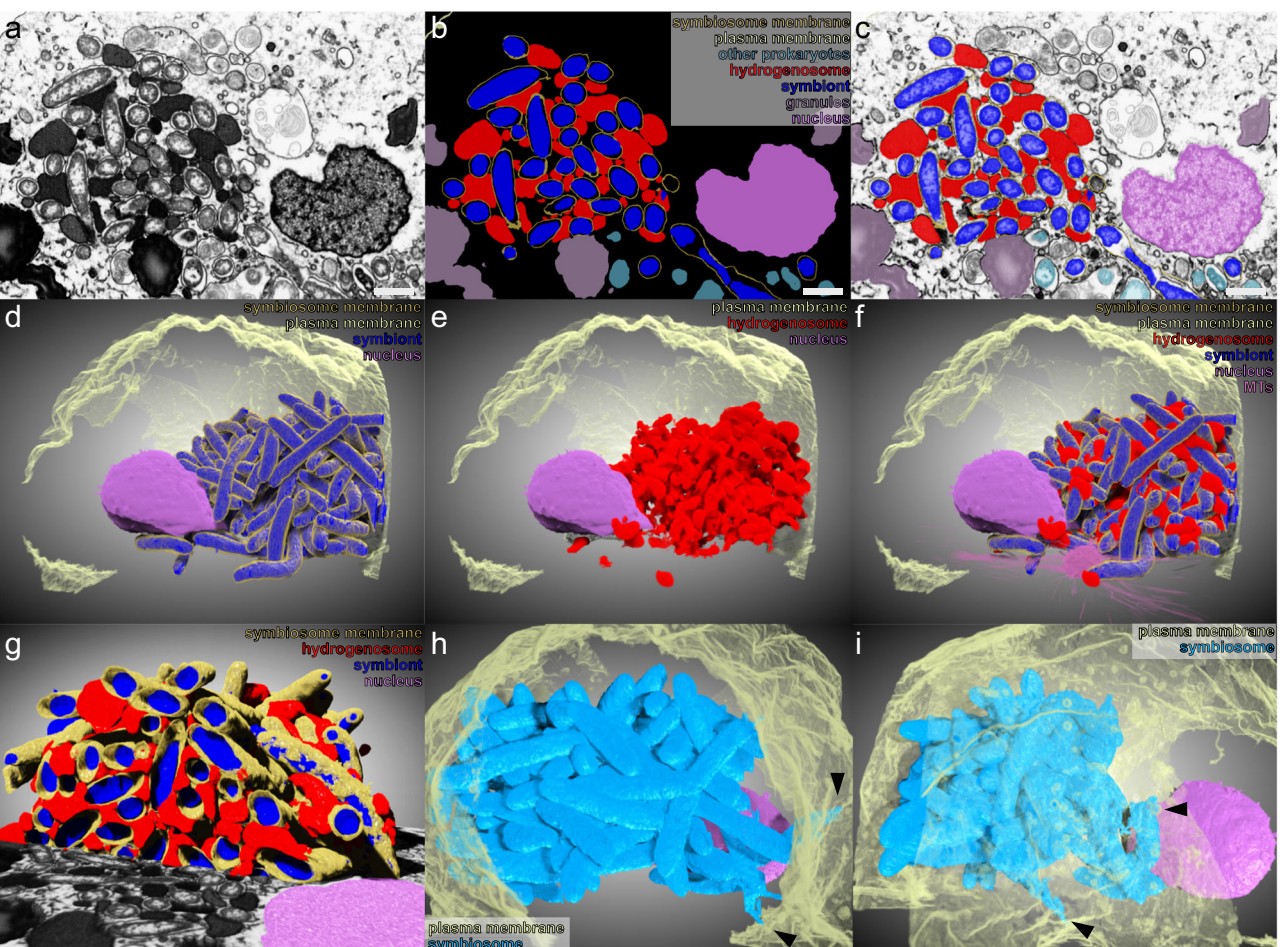

**Fig. 2 | *Anaeramoeba* symbionts are housed in a membrane network with connections to the plasma membrane. a–c** FIB-SEM slice of *A. flamelloides* BUS-SELTON2, showing (**b**) segmented regions of interest (symbiont–blue, hydrogenosome–red, symbiosome membrane–gold, nucleus–purple, granules–aubergine, other prokaryotes– blue green, plasma membrane–yellow). **c** overlay of (**a**, **b**). **d–f** Rendered volume of FIB-SEM slices showing segmented regions of interest. **d** Symbiont, symbiosome membrane, nucleus and plasma membrane. **e** Hydrogenosomes, nucleus and plasma membrane. **f** Overlay of all regions of interest in (**d**, **e**), and microtubules (MTs) (symbiont–blue, hydrogenosome–red, symbiosome membrane–gold, nucleus–purple, plasma membrane– yellow, MTs–pink). **g** A clipped 3D rendering of *A. flamelloides* BUSSELTON2 symbionts (blue), hydrogenosomes (red), symbiosome membrane (gold) and nucleus (purple) showing the internal structure of the symbiosome. **h–i** Two different views of 3D surface rendering of the components of the sym-biosome connected to the plasma membrane (light blue). Plasma membrane (yellow) and nucleus (purple). Symbiosome connections to the plasma membrane are indicated by black arrows. Scale bar (**a–c**), 1 μm.

## Results

### The *Anaeramoeba* symbionts are directly connected to the extracellular milieu

Actively feeding *Anaeramoeba* cells have a fan-shaped hyaline zone and trailing projections (Fig. 1a). Near the nucleus in the bulbous cell body lies a large mass of symbionts housed in compartments with single bounding membranes tightly positioned close to, but not in direct contact with, hydrogenosomes (Fig. 1b–e)[6,7]. *Anaeramoeba* cells have an acentriolar centrosome positioned below the hydrogenosome-symbiont mass and the nucleus from which micro-tubules radiate throughout the ventral side of the cell (Fig. 1b, e). The symbionts are stably maintained throughout the cell cycle and segre-gated in an organized fashion during cell division (Fig. S1). These observations indicate that the symbionts are housed in stable struc-tures but do not reveal if the *Anaeramoeba* symbionts are endo-symbionts completely enclosed within the host or are directly connected to the extracellular media.

To distinguish between these two possibilities, we first used live-cell pulse-labeling experiments with fluorescent-labeled Wheat Germ Agglutinin (WGA), a lectin that stains sialic acid and N-acetyl glucosa-mine in bacterial cell walls. The *Anaeramoeba flamelloides*

BUSSELTON2 symbiont showed clear WGA labeling after only 10 min of incubation, with staining intensity comparable to free-living bacteria (Fig. S2). This suggests that the symbionts have relatively quick "access" to the extracellular environment. Furthermore, transmission electron microscopy images of *Anaeramoeba ignava* BMAN show symbiont cells in pockets at the host cell membrane with connections to the surface (Fig. 1f).

To better resolve the symbiosome structure, we conducted FIB-SEM tomography on *A. flamelloides* BUSSELTON2 holobionts. After 1780 serial section SEM images were collected in 7 nm deep incre-ments through a fixed cell, a computational 3D reconstruction was performed with specific distinctions made between symbionts, other prokaryotes, hydrogenosomes, symbiosome-membrane, micro-tubules and the acentriolar centrosome, plasma membrane, the nucleus, and dense granules (Figs. 2 and S3). The 3D reconstructions show the mass of symbionts and the symbiosome structure housing them (Fig. 2d) is highly elaborate, tightly associated with the hydro-genosomes (Fig. 2e–g) (Movie S1), and located proximal to the nucleus. Reconstruction of the plasma membrane alongside the symbiosome revealed membrane connections between symbiosome compartments and the plasma membrane (arrowheads in Fig. 2h, i).

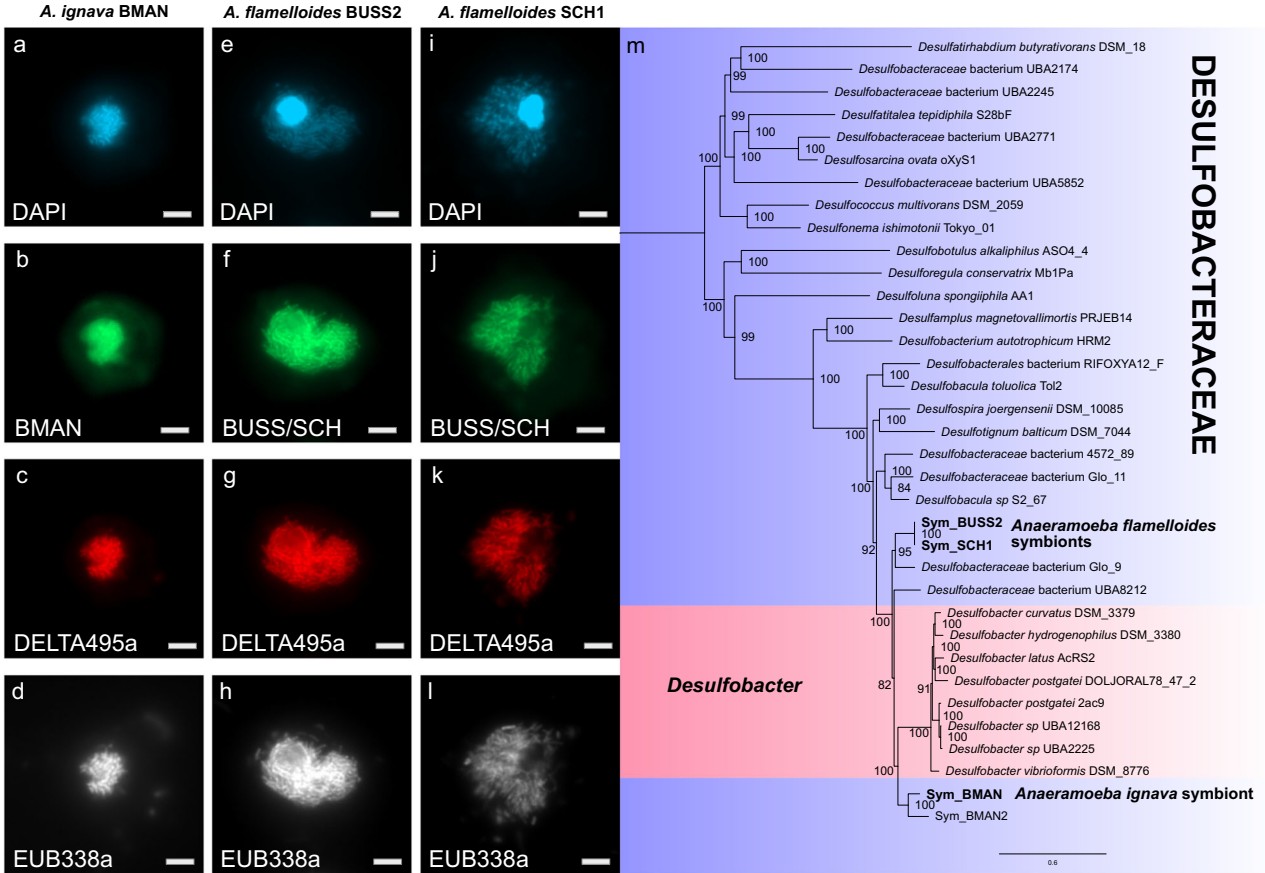

**Fig. 3 | *Anaeramoeba* symbionts are closely related to *Desulfobacter* and acquired in separate events.** *A. ignava* BMAN stained with (**a**), DAPI and hybridized with (**b**), probe DSBA355-BMN-488, (**c**), probe Delta495a-Atto 550 and (**d**), probe EUB338a-Atto 633. *A. flamelloides* BUSSELTON2 stained with (**e**), DAPI and hybridized with (**f**), probe BUSS/SCH-BMN-488, (**g**), probe Delta495a-Atto 550 and (**h**), probe EUB338a-Atto 633. *A. flamelloides* SCHOONER1 stained with (**i**), DAPI and hybridized with (**j**), probe BUSS/SCH-BMN-488, (**k**), probe Delta495a-Atto 550 and (**l**), probe EUB338a-Atto 633. **m** The phylogenomic analysis was based on 36 taxa, 108 proteins, and 25,918 sites with IQTree v2.2.0.3 (LG + C60 + F + G model of evolution). Bipartition support values are derived from 100 non-parametric bootstraps under the PMSF model. Scale bar indicates inferred number of substitutions per site. Tree files and alignments are available at FigShare: https://doi.org/10.6084/m9.figshare.20375619. Scale bar (**a**–**l**), 1 μm. The experiments in (**a**–**d**), and (**e**–**l**), show representative images from duplicate and triplicate hybridizations respectively.

Analyses of the internal structure of the symbiosome revealed that the various compartments housing distinct symbionts (indicated in gold in Fig. 2g) are connected by tubular channels (Fig. S4). We observed 171 individual membrane connections between symbiosome subcompartments yielding fifteen connected components with two or more symbionts (Fig. S5A–H and Movie S2). In total, we found that in the specific *Anaeramoeba* cell analyzed, 86% of symbionts (158/183) have contact with at least one other symbiont while 25 symbionts are housed in individual compartments (Fig. S5 and Movie S2). The largest symbiosome component harbors 105 symbionts (Fig. S5i–l). Collectively, 108 symbionts are directly in contact with the outside media (Fig. 2h, i and S5). Even though the 25 individual compartments tend to be peripheral, it should be noted that they make very close approaches to the multi-symbiont compartments (Fig. S5 and Movie S2).

### *Anaeramoeba* symbionts belong to Desulfobacteraceae and were acquired independently in different host species

We sequenced the nuclear genomes of three *Anaerameoba* isolates (*A. ignava* BMAN, *A. flamelloides* BUSSELTON2, and SCHOONER1) and the genomes of the associated prokaryotes using Nanopore long-read and Illumina short-read technologies (Table S1). The most abundant prokaryotes in the *A. flamelloides* and *A. ignava* genomic datasets were Desulfobacteraceae, hereafter referred to as Sym_BUSS2, Sym_SCH1,

and Sym_BMAN. In *A. flamelloides*, where ameba separation from suspended bacteria was found to be the most efficient (Fig. S6), the symbiont lineages were detected almost exclusively in the amoeba enriched fraction and were virtually non-detectable in the culture supernatant (Fig. S6). Fluorescence in situ hybridization (FISH) using unique probes designed against the 16S rRNA genes in the genomes of Sym_BMAN (Fig. 3a–d) and Sym_BUSS2/Sym_SCH1 (Figs. 3f–l and S7) (as well as less specific probes for broader encompassing taxonomic groups (Fig. S8)) confirmed the identity of the symbionts. The average number of symbionts per host based on symbiont genome coverage relative to nuclear genome coverage was estimated to be 35.3–36.5 for *A. flamelloides* (Sym_BUSS2 & Sym_SCH1) and 12.6 for *A. ignava* (Sym_BMAN) (the ploidy of the host nuclear genome is unknown but for these analyses posited to be haploid). All three symbiont genomes are similar in size to free-living relatives (4.97–6.06 Mbp) and relatively gene-rich (3823–5286 intact genes) (Table S2). The Sym_BUSS2 and Sym_SCH1 genomes are 99.7% identical at the nucleotide level but show large differences in synteny (Fig. S9), whereas the Sym_BMAN genome is highly divergent relative to the two *A. flamelloides* symbionts (average nucleotide identity of 74.1–74.2%). A second less abundant Desulfobacteraceae genome, Sym_BMAN2, was also detected in the *A. ignava* dataset but its status as a symbiont could not be established using FISH. Thus, for *A. ignava* we focused on Sym_BMAN in subsequent analyses.

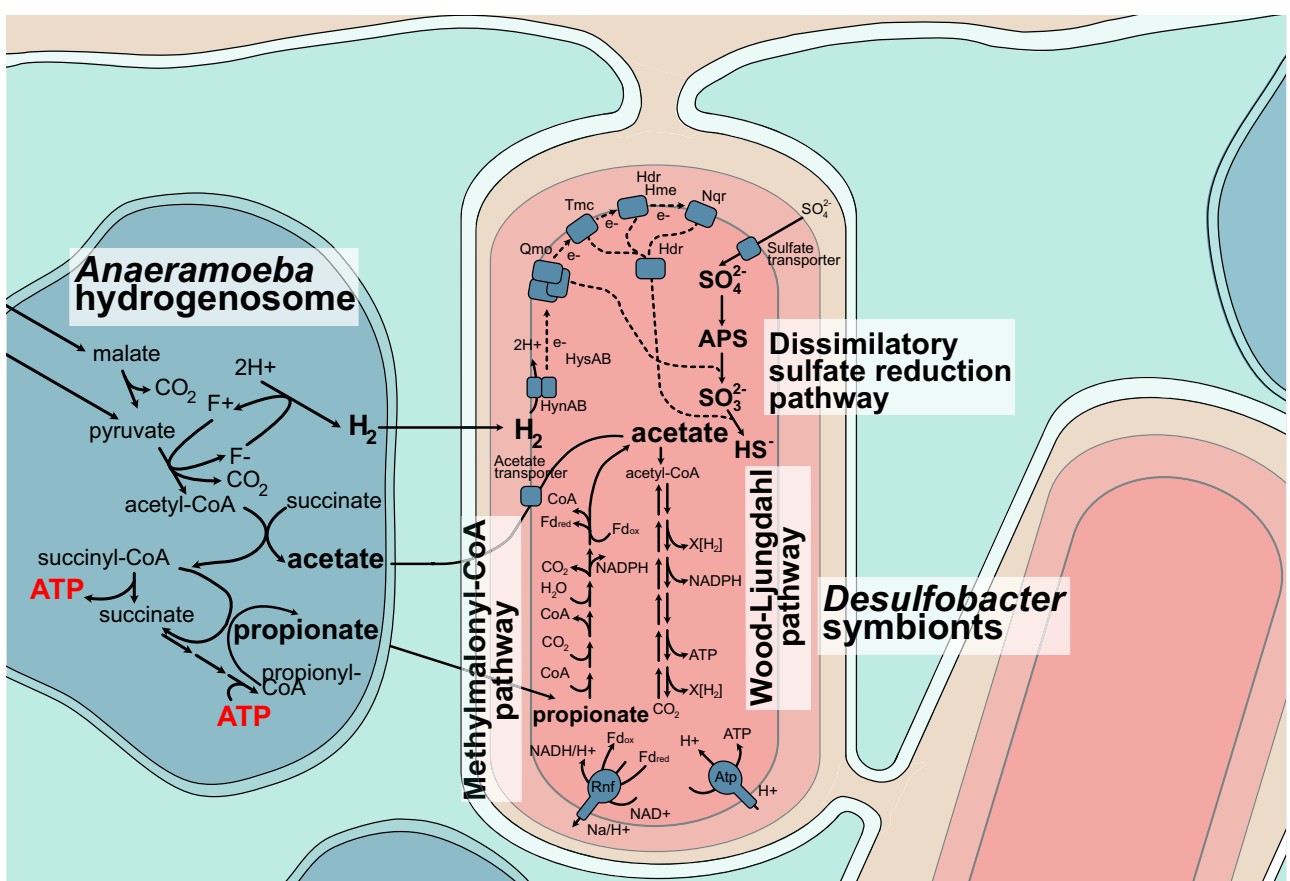

**Fig. 4 | *Anaeramoeba* symbionts are metabolically poised for syntrophic interaction with hydrogenosomes.** Suggested syntrophic interactions between *Anaeramoeba* hydrogenosomes (blue) and symbionts (pink/salmon) based on metabolic reconstruction from transcriptomic and genomic evidence. The ATP-producing hydrogenosomes generate $H_2$, acetate, and propionate as end-products (in bold). Based on metatranscriptomic data, the symbionts use the hydrogenosome products by prominently expressing the dissimilatory sulfate reduction (DSR), methylmalonyl-CoA, and the Wood-Ljungdahl pathways. The symbionts are in deep membrane-pits with a connection to the cell surrounding that gives ready access to sulfate (gold). HynAB periplasmic [NiFe] hydrogenase, HysAB [NiFeSe] hydrogenase, Qmo QmoABC complex, Tmc TmcABCD complex, Hdr Hetero-disulfide reductase, Hme DsrMKJOP complex, Nqr NADH:ubiquinone oxidor-eductase, Rnf Rnf complex, Atp ATP synthase, APS adenosine 5′-phosphosulfate, CoA Coenzyme A, NAD Nicotinamide adenine dinucleotide, NADPH Nicotinamide adenine dinucleotide phosphate, $Fd_{red/ox}$ Ferredoxin reduced/oxidized, ATP adenosine triphosphate.

Phylogenetic analysis of 15 ribosomal proteins from 165 Desulfo-bacterales genomes placed the symbionts as members of the family Desulfobacteraceae (Fig. S10), closely related to the genus *Desulfo-bacter*. To improve resolution, we performed phylogenetic analysis on a data set of 108 proteins from a phylogenetically restricted set of Desulfobacteraceae. Sym_BMAN and Sym_BMAN2 belong to a clade that branches sister to a large *Desulfobacter* group (Fig. 3m) whereas Sym_BUSS2/SCH1 branches separately, forming a well-supported group (BS 95%) with an environmental isolate (Glo_9) recovered from a metagenome of the benthic foraminiferan *Globobulimina* spp. The latter clade forms a sister group to the *Desulfobacter* – Sym_BMAN clade. The *Desulfobacter*-like symbionts of *A. ignava* and *A. flamelloides* thus appear to have been acquired independently by their respective hosts.

## The *Anaeramoeba-Desulfobacter* symbiosis was recently established

The Sym_BUSS2 and Sym_SCH1 genomes were found to have 922 and 1000 pseudogenes, respectively (Table S2 and Supplementary Data 1), and >600 Insertion Sequences (IS) per genome (Supplementary Data 2). In contrast, pseudogene and IS element abundance in Sym_BMAN is an order of magnitude lower and falls within the range of free-living Desulfobacteraceae species (Table S2). The IS elements of Sym_BUSS2 and Sym_SCH1, many of which have an intact transposase and are likely active, showed no strong evidence of clustering (Fig. S11A, B). However, >200 IS elements were found to be close (<1500 bp) to the edges of syntenic blocks, suggesting that the apparent high frequency of rearrangements is connected to IS element activity (Fig. S11C, D). The Sym_BUSS2 and Sym_SCH1 genomes are enriched in pseudogenes with predicted gene ontologies related to signal transduction (T) and amino acid metabolism and transport (E) functional categories, whereas translation (J) and transcription (K) classes showed the opposite trend (Fig. S12). The genomes of these two symbionts have flagellar operons (Fig. S13), type IV pili, and CRISPR systems that are all extensively pseudogenized; both sym-bionts also appear to be impaired in their ability to decorate lipid A with O-antigen. In contrast, the Sym_BMAN genome encodes an intact type I-F CRISPR system with an array of 25 spacers as well as two convergent, 25 gene operons for a type VI secretion system (T6SS) (Fig. S14). The greater degree of genome degeneration seen in the Sym_BUSS2 and Sym_SCH1 genomes relative to the largely intact Sym_BMAN genome strongly suggests that the former are evolu-tionarily "older" symbionts and that they are unable to survive without the host.

However, the relatively large genomes of all *Anaeramoeba* sym-bionts and other genomic features are reminiscent of early-stage, recently-acquired symbionts[8]. Constitutive overexpression of the chaperonin GroEL/S is regarded as a critical factor in stabilizing endosymbionts in a wide range of insect symbioses[9]. Notably, meta-transcriptomic analysis of the *A. ignava* BMAN and *A. flamelloides*

**Table 1 | Predicted LGTs in the genomes of *Anaeramoeba* species**

| | Predicted proteins | Ortho groups (OGs) | OGs including at least one prokaryotic or viral sequence | Trees[a] | Inferred LGT events | Total LGT-derived genes |
|---|---|---|---|---|---|---|
| *A. ignava* BMAN | 15,022 | 6168 | 2224 | 1743 | 388 | 612 |
| *A. flamelloides* BUSSELTON2 | 29,927 | 14,259 | 3701 | 3156 | 781 | 1414 |
| *A. flamelloides* SCHOONER1 | 30,041 | 14,246 | 3679 | 3115 | 777 | 1359 |

[a]Trees were estimated for only a subset of OGs that contained at least one prokaryotic or viral taxa and with >80 sites (see "Methods")

BUSSELTON2 cultures show that the *groEL, groES,* and HSP70-encoding *dnaK* genes are highly expressed in Sym_BUSS2 but moderately to lowly expressed in Sym_BMAN, in line with their respective degrees of genome erosion (Supplementary Data 3).

### *Anaeramoeba* symbionts are metabolically poised to use hydrogenosomal metabolites

The hydrogenosomes of both *Anaeramoeba* species are predicted to produce $H_2$, acetate, and propionate as end-products[6]. To investigate whether the symbionts express genes predicted to be important for $H_2$-uptake, we performed metatranscriptomics of the *A. ignava* BMAN and *A. flamelloides* BUSSELTON2 cultures. Metatranscriptomics of Sym_BUSS2 showed that the most prominently symbiont-expressed genes include those involved in dissimilatory sulfate reduction (DSR), the group 1b uptake [Ni/Fe] hydrogenase (*hynAB*), the Wood–Ljungdahl (W–L) pathway, and ATP synthase (Fig. 4 and Supplementary Data 3). Similarly, in Sym_BMAN, the *dsr* genes, *aprAB* pathway and *hynAB* were highly expressed, while W–L pathway and ATP synthase genes are less expressed than in Sym_BUSS2 (Supplementary Data 3). Sym_BMAN shows high expression of an acetate permease (*actP*) that might act to bring in host-derived acetate to feed the W–L pathway. Although we could not identify acetate transporter genes (*actP* or *satP*) in the *A. flamelloides* symbiont genomes, we predict these symbionts might be able to acquire acetate by diffusion across the membrane as reported in other systems[10]. The symbionts appear to be able to activate propionate via propionyl-CoA synthase (*prpE*), which is highly expressed in Sym_BUSS2 and Sym_BMAN (Supplementary Data 3). Propionyl-CoA likely feeds into the methylmalonyl-CoA (MMA) pathway to produce pyruvate that, via oxidative decarboxylation by pyruvate:ferredoxin oxidoreductase, generates acetyl-CoA that could enter the W–L pathway (Fig. 4). Most enzymes of the MMA pathway are highly and moderately expressed in Sym_BUSS2 and Sym_BMAN, respectively. Because Sym_BUSS2 is closely related to the *Desulfobacter* Glo_9 denitrifying symbiont of *Globobulimina* spp. (discussed in refs. 11,12), we investigated whether the *Anaeramoeba*:Sym_BUSS2 symbiotic system could be based on denitrification. However, we failed to find denitrification-related genes in the host (*nrt, nirK, nor*) or symbionts (*napA, nozA*).

Given that many of the gene products described above are sensitive to oxygen, we examined the genomes of the symbionts for oxygen or reactive oxygen species (ROS) defense systems. Genes encoding the superoxide detoxification (superoxide reductase and rubredoxin) and hydrogen peroxide detoxification (rubrerythrin) were highly expressed by Sym_BUSS2 and Sym_BMAN (Supplementary Data 3).

### *Anaeramoeba* nuclear genomes are A + T rich and have greatly expanded gene families

The nuclear genomes of Anaeramoebae are A + T-rich, with the *A. ignava* BMAN nuclear genome (80.65% A + T) and intron sequences (91.86% A + T) being among the highest ever reported in eukaryotes (e.g., the *Plasmodium falciparum* nuclear genome and intron sequences are 80.6% and 86.5% A + T, respectively[13]) (Tables S1 and S3A).

Whereas the assembled genomes of *A. flamelloides* are almost six-times larger than for *A. ignava*, the number of protein coding genes is only two-fold higher (Table S1). The sequence divergence between the *Anaeramoeba* species is substantial with only 37.3% average amino acid identity between orthologous proteins (Fig. S15). In contrast, the *A. flamelloides* BUSSELTON2 and SCHOONER1 strains are much more similar (94.8% identity between orthologs).

To investigate the evolution of gene content in Anaeramoebae, we compared the gene families of the three *Anaeramoeba* genomes to nine other microbial eukaryotes including species from Parabasalia, Oxymonadida, Fornicata, Heterolobosea and Amoebozoa (Supplementary Data 4). Overall 565 and 4704 families were deemed core and accessory, respectively (Supplementary Data 4A–C). Of the accessory families 67, 74 and 107 are specific to BUSSELTON2, SCHOONER1 and BMAN, respectively (Supplementary Data 4D) with 92 families uniquely found in all three of them (Supplementary Data 4E). When comparing only *A. flamelloides* to *A. ignava*, expansions were detected in 330 shared (core to Anaeramoebae) families while contractions were found in 13 such families (Supplementary Data 4H). The contractions seen in *A. flamelloides* likely correspond to expansions in *A. ignava* rather than contractions in *A. flamelloides* (Supplementary Data 4F–H). Together with the presence of BUSSELTON2, SCHOONER1 and BMAN-specific families already mentioned, these differences highlight the striking differences among these relatively closely related taxa.

Many gene family expansions occur specifically in *A. flamelloides* and are enriched in genes involved in RNA and DNA metabolism and membrane-trafficking systems. One striking feature is the massive expansion of ribosomal proteins with more than 1100 genes each in *A. flamelloides* compared to the 82 genes in *A. ignava*, representing a more typical number for a eukaryote (Supplementary Note 1, Supplementary Data 4I–K and Fig. S16). However, expansions of gene families connected to nutrient exchange were also noted. For example, *A. flamelloides* and *A. ignava* have sizable expansions of an Amt/MEP ammonium transporter gene, whose products likely function in the uptake or secretion of ammonia, a primary nitrogen source for cells[14].

### Lateral gene transfer is ongoing in *Anaeramoeba*

LGT is an important mechanism impacting genome evolution in many eukaryotes[15,16]. We performed a phylogenomic analyses to detect LGTs from prokaryotes and viruses to *Anaeramoeba*. We identified 612, 1414, and 1359 putative foreign genes in the BMAN, BUSSELTON2, and SCHOONER1 genomes, respectively (4.1%, 4.7%, 4.5% of the total number of genes per genome). Accounting for gene duplications after acquisition, this corresponds to 388, 781, and 777 gene families (Table 1). These are some of the highest LGT proportions ever reported for protists[17]. Many of the LGTs appear to have happened after the divergence of *A. ignava* and *A. flamelloides* (Supplementary Data 5 and Fig. 5). 169 putative LGTs occurred in a common ancestor of the two species, whereas 577 LGTs map to the base of the divergence between the *A. flamelloides* BUSSELTON2 and SCHOONER1 strains, and 220 were acquired along the lineage leading to *A. ignava* (this pattern may also be explained in part by differential loss of LGT genes). Over 30

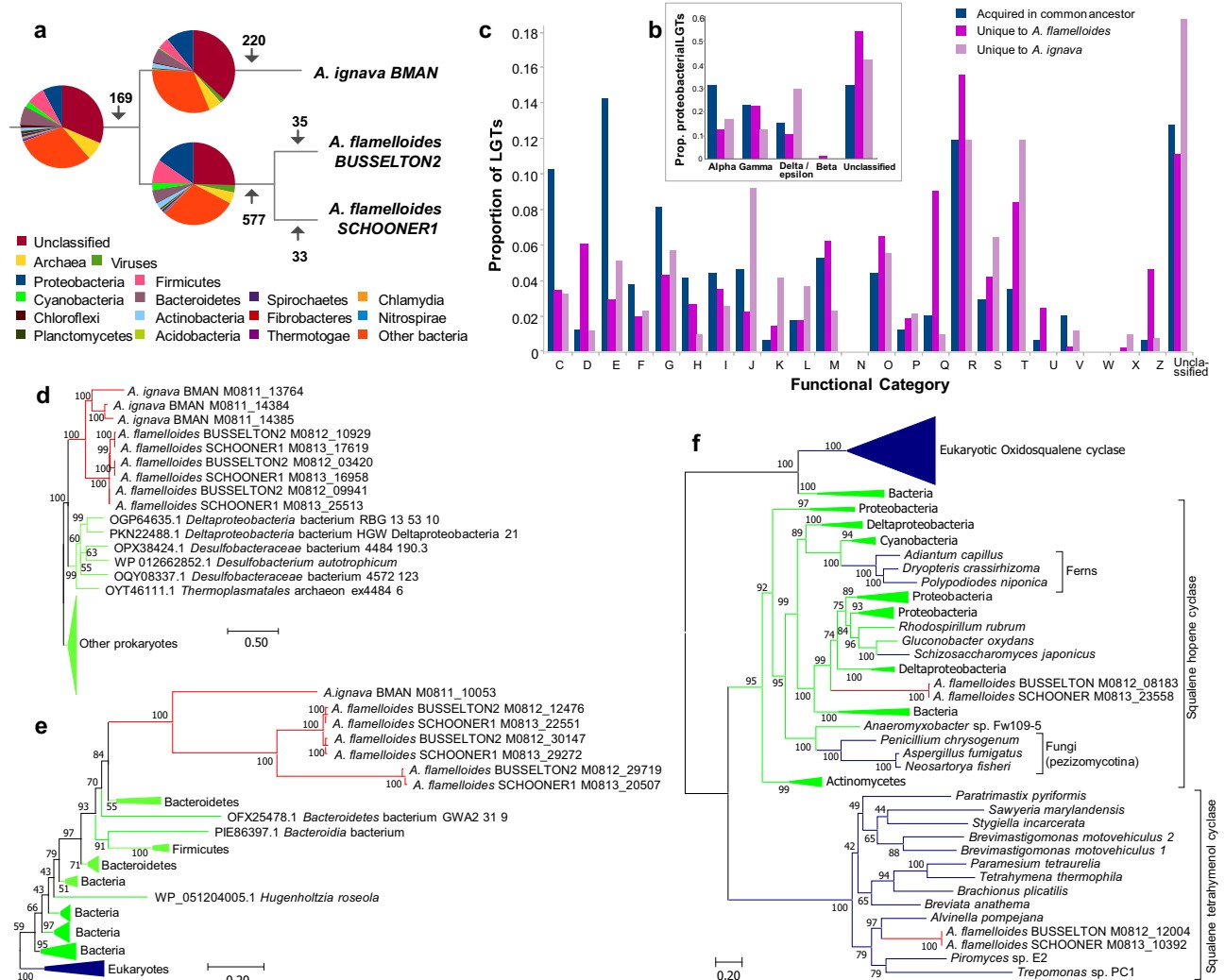

**Fig. 5 | Taxonomic range and functional categories of LGT donors in *Anaeramoeba*. a** Taxonomy of LGT donors for genes inferred to have been acquired in the common ancestor of *Anaeramoeba* and separately in *A. flamelloides* and *A. ignava*. The numbers on the branches are the inferred number of LGTs on each position of the tree. **b** Breakdown of LGTs from proteobacterial donors. **c** Functional categories of genes inferred to have been acquired in the *Anaeramoeba* common ancestor, in *A. flamelloides* and *A. ignava*. Color coding is the same as for part (**b**). **d** Maximum-likelihood phylogeny of FAD-dependent oxidoreductases. **e** Maximum-likelihood phylogeny of vitamin B12-dependent methionine synthase MetH. The trees shown in (**d**) and (**e**) were produced by our LGT-detection pipeline (see text). **f** Maximum-likelihood phylogeny of oxidosqualene cyclase (OSC), squalene-hopene cyclase (SHC), and squalene-tetrahymanol cyclase (STC). The tree shown stems from analyses based on previously published datasets[84,85]. Sequences were aligned with MAFFT, sites were selected using BMGE, and the phylogeny inferred using IQTree model C20 + G4 with 1000 ultrafast bootstraps. *Anaeramoeba* sequences are in red, eukaryotic sequences in blue, and prokaryotic

sequences in bright green. Scale bars indicate the inferred number of amino acid substitutions per site. Abbreviations: INFORMATION STORAGE AND PROCESSING: [J] Translation, ribosomal structure and biogenesis; [K] Transcription; [L] Replication, recombination and repair; CELLULAR PROCESSES AND SIGNALING: [D] Cell cycle control, cell division, chromosome partitioning; [V] Defense mechanisms; [T] Signal transduction mechanisms; [M] Cell wall/membrane/envelope biogenesis; [N] Cell motility; [Z] Cytoskeleton; [W] Extracellular structures; [U] Intracellular trafficking, secretion, and vesicular transport; [O] Posttranslational modification, protein turnover, chaperones. METABOLISM: [C] Energy production and conversion; [G] Carbohydrate transport and metabolism; [E] Amino acid transport and metabolism; [F] Nucleotide transport and metabolism; [H] Coenzyme transport and metabolism; [I] Lipid transport and metabolism; [P] Inorganic ion transport and metabolism; [Q] Secondary metabolites biosynthesis, transport and catabolism. POORLY CHARACTERIZED: [R] General function prediction only; [X] Mobilome: prophages, transposons; [S] Function unknown.

genes were inferred to have been acquired in the two *A. flamelloides* strains after their divergence.

Most of the LGTs for which a taxonomic origin could be inferred are bacterial, although archaeal and viral contributions to *Anaeramoeba* are also apparent (Fig. 5a). Proteobacteria, Firmicutes and Bacteroidetes (Fig. 5a) are the most represented donor phyla contributing genes to the ancestor of *A. ignava* and *A. flamelloides*, and the two species appear to have acquired more proteobacterial genes since they diverged from one another. Interestingly, Deltaproteobacteria is not the most represented class, despite the taxonomic affinity of

*Anaeramoeba*'s symbionts (Fig. 3). That said, several dozen genes with amino acid identities >60% (Supplementary Data 5D) are inferred to have been transferred from Deltaproteobacteria to *Anaeramoeba*, such as an aspartate ammonia ligase (in *A. flamelloides*) and an FAD-dependent oxidoreductase (in both species) (Fig. 5d). A few *Anaeramoeba* genes have discernable homologs only in the symbionts and thus might represent symbiont-to-host LGTs. Curiously, these LGTs encode long repetitive proteins (up to 4100 amino acids). The relative contributions of Alpha- and Gammaproteobacteria are similar for *A. ignava* and *A. flamelloides* (Fig. 5b).

## Laterally transferred genes shape *Anaeramoeba* biology

The predicted functions of LGTs in *Anaeramoeba* differ between those acquired in the common ancestor of *A. ignava* and *A. flamelloides* and those acquired after the divergence of the two species (Fig. 5c and Supplementary Note 2). Several genes acquired in the *Anaeramoeba* common ancestor may be related to accommodating sulfate-reducing symbionts. For example, genes encoding a cysteine synthase K and D-3-phosphoglycerate dehydrogenase are LGT-derived (Fig. S17D, E); together with SerC phosphoserine aminotransferase (which was not detected in our LGT screen but was found in the genomes), these proteins link glycolysis to the recycling of SH- produced by the symbionts and the generation of acetate that might be used by the symbiont (Fig. S18). A prokaryotic acetate transporter gene (Fig. S17F) also appears to have been acquired in the *Anaeramoeba* ancestor and might have played a role in the establishment of the symbiosis (the symbionts are predicted to use acetate produced by *Anaeramoeba*). The foreign acetate transporter gene is present in >10 copies in each *Anaeramoeba* genome.

We scanned the *Anaeramoeba* LGTs for putative host-symbiont recognition factors, which often have repetitive domains[18]. Amongst the LGT candidates, we identified 38 gene families that previously have been associated with host-symbiont interactions (Supplementary Data 5). Some of these gene families are highly amplified, have predicted signal peptide-encoding regions, and show differential presence/abundance between *A. ignava* and *A. flamelloides*, indicating that they might traffic in the secretory pathway and could potentially mediate symbiont interactions. Several foreign genes in *A. ignava* BMAN and *A. flamelloides* BUSSELTON2 are clearly involved in anaerobic life, including (i) MRO metabolism[6], (see Supplementary Data 5), (ii) oxygen detoxification, (iii) ATP production in the absence of oxidative phosphorylation (e.g., pyruvate phosphate dikinase; Fig. S17G), and (iv) cell membrane composition modification (acquisition of a bacterial squalene hopene cyclase (SHC) involved in oxygen-free biosynthesis of hopanoids (Supplementary Note 2 and Fig. 5f)). Several of these genes appear to have been acquired on multiple occasions from various donors (Supplementary Note 2, e.g., Fig. S17M−O).

Of the LGT-derived genes that are amenable to functional/metabolic prediction, very few make up a complete "module"[19], suggesting that most LGT-derived proteins function in concert with host-origin enzymes in mosaic pathways. The sole exception is a complete set of genes encoding the enzymes of the Leloir pathway for galactose metabolism in *A. flamelloides* (see Supplementary Note 2).

### *Anaeramoeba* might acquire vitamin B12 from its symbionts

Vitamin (vit)B12 is one of the most complex coenzymes known. Its biosynthesis involves ~30 enzymatic steps and is confined to certain bacterial and archaeal species[20]. The symbionts associated with both *Anaeramoeba* species encode a complete anaerobic pathway for vitB12 synthesis (except for alpha-ribazol phosphatase (CobC); below). Interestingly, our genome screens detected up to six vitB12-dependent enzymes encoded in the *Anaeramoeba* host genomes, a surprising number and a record among eukaryotes examined thus far. This includes two enzymes in *A. flamelloides* (Supplementary Note 3) that have not been found in a eukaryote before. Additionally, both *Anaeramoeba* species have acquired a class II ribonucleotide reductase (RNR) and a methionine synthase (*metH*) (Fig. 5E) from bacteria and have a eukaryotic methylmalonyl-CoA mutase and its associated GTPase MeaB. These observations suggest that vitB12−or cobalamin− is produced by the symbionts and consumed by the host (both *Anaeramoeba* species encode a cobalamin adenosyl transferase that catalyzes the conversion of cobalamin into adenosylcobalamin, the form used as a cofactor by vitB12-dependent methylmalonyl-CoA mutase). The exchange of vitB12 between prokaryotic producers and B12-auxotrophic algae was hypothesized elsewhere[21,22]. Curiously, we found a laterally acquired gene encoding CobC (Fig. S17W) in *A. ignava*.

This enzyme catalyzes the last step of cobalamin synthesis, which is not predicted to occur in the symbionts. The *cobC* gene has also been acquired in some vitB12 auxotrophic diatoms[23]. Our analyses strongly suggest that vitB12 exchange is part of the host-symbiont metabolic interaction in *Anaeramoeba*, a hypothesis that can be tested in the future.

### Expansions of endosome/phagosome modulating membrane-trafficking proteins in *Anaeramoeba*

The vesicle-bound symbiont mass in these Anaeramoebae, which is connected to the extracellular environment, is reminiscent of phagosomes that have been delayed or frozen in their maturation process and then elaborated. The general expansions in genes involved in membrane-trafficking systems in both *Anaeramoeba* spp. identified above prompted us to specifically investigate several sets of proteins, which in other eukaryotes have been implicated in regulation of phagosomal maturation. Certain Rab GTPases are essential in this process, mediating early to late endosomal conversion, as well as endosomal function and are involved in other processes such as signal transduction and acting as molecular switches that regulate formation and transportation of vesicles[24,25]. Consequently, we investigated not only Rabs but also their GTPase Activating Proteins (GAPs) and the Vps-C complexes HOPS and CORVET that mediate endosomal Rab conversion, with the former having recently been implicated in regulating phagosomal-lysosomal fusion[26].

The *Anaeramoeba* genomes encode many Rab GTPases, almost 400 in *A. flamelloides*. Rabs are small proteins, which made resolved classification by phylogenetic analysis intractable. Orthofinder[27] was thus used for initial classification followed by targeted phylogenies of related Rab sub-families (when necessary). While 75−80% of Rab proteins could not be confidently classified, clear trends were nevertheless apparent (Supplementary Data 6). We observed relatively low numbers of Rabs 6, 7, and 18, and no clear candidate orthologs for eight Rab sub-families were found, including some endosomal Rabs such as Rab 4 or 22. By contrast, Rab GTPase families 1, 2, 5, 8, 11, 14, 20, 21, and 24 were found to be highly expanded (Fig. S19A−C and Supplementary Data 6). We were also able to confidently classify most Tre-2/Bub2/Cdc16 (TBC) proteins that serve as GAPs for Rabs. Although there are only one or several members of each of the TBC-I, M, and N subfamilies, the TBC-E, F, K, and Q subfamilies are notably expanded (Fig. S19D). Finally, we observed expansions in the complement of both HOPS and CORVET subunits, most obviously with increased numbers of the HOPS-specific subunits (Supplementary Data 6).

We searched for evidence of gene duplications within the Rab and TBC subfamilies that might predate the divergence of the *Anaeramoeba* spp., and which could indicate *Anaeramoeba*-specific machinery involved in capture and regulated retention/digestion of symbionts. Rab1, or the metazoan-specific paralog Rab35[28], has been directly implicated in phagosome maturation dynamics in mammalian cells[29] and in the amoebozoan parasite *Entamoeba*[30]. Although the phylogenetic resolution between the subfamilies is poor, in the case of the Rab1 sub-family (Fig. S19C), and its regulator TBC-E (Fig. S19D), tree topologies are consistent with gene duplication and diversification prior to the split of the *Anaeramoeba* spp. We also observed three strongly supported clades within TBC-K (Fig. S19D) that predate the *Anaeramoeba* split. TBC-K regulates another GTPase Arf6[31], which was recently shown in metazoan systems to regulate phagosomal maturation. Notably, Rab35 and Arf6 operate in a mutually antagonistic mechanism in multiple mammalian systems[32,33].

Collectively, these analyses show expansions of the complement of membrane-trafficking components in *Anaeramoeba* spp. that, in other organisms, regulate phagosomal maturation. But this family complement expansion is not true of all *Anaeramoeba* membrane-trafficking machinery. A recent study[34] of the vesicle coat forming machinery (including coats that act within the endosomal system)

found these proteins to be largely encoded by a single copy in *A. ignava*, resembling the canonical eukaryotic complement. Overall, our findings are consistent with the existence of a specialized modulation and control system for the symbiont-containing vacuoles that allows *Anaeramoeba* spp. to regulate the acquisition and maintenance of its symbiont mass.

## Discussion

Symbiosomes have evolved on numerous occasions across the tree of life and their origins are associated with massive changes to host cell biology and physiology[1]. Our analyses suggest that the ancestor of *Anaeramoeba* spp. evolved a subcellular symbiosome to house *Desulfobacter*-related symbionts in a way that has drastically reshaped their cell biology and metabolic capacities. The notable expansion of some membrane-trafficking genes in the *Anaeramoeba* common ancestor suggests that the membrane compartment housing the symbionts is a highly evolved structure that might allow them to selectively manage their captured symbionts and position them in tight association with their hydrogenosomes. To our knowledge, the only superficially similar subcellular arrangements of hydrogenosomes and symbionts have been observed in certain anaerobic ciliates (Scuticociliatia and Karyorelictea)[4,5], the heterolobosean *Psalteriomonas lanterna*[35], and obligately symbiotic parabasalids (*Trichonympha* and *Spirotrichonympha*), which are found in the termite digestive tract[36,37]. These organisms appear to have metabolic interdependencies with endosymbiotic and ectosymbiotic prokaryotes—often methanogens and/or sulfate-reducing bacteria—with some of the latter being housed in shallow host-derived cell membrane invaginations with hydrogenosomes in close proximity[36,37]. While the symbionts in *Anaeramoeba* are almost completely compartmentalized, they nevertheless retain connections to the external environment (Fig. 2h, i), presumably due to metabolic constraints such as the need to access sulfate, as seen in the *Desulfovibrio* symbionts of *Trichonympha*[37,38]. Brown-pigmented karyorelictid ciliates in anoxic sediments house several symbiont consortia including a member of Desulfobacteraceae that are not closely related to the Anaeramoebae symbionts (Fig. S10). These ciliate symbionts heavily transcribe genes needed for DSR even though they appear to reside in double-membraned host-derived vacuoles with no obvious mechanism for sulfate provisioning[5]. In the Anaeramoebae, it remains unclear the degree to which the cell surface connections to the symbiosomes are relatively static or continuously forming and closing. Regardless, given the many fine connections between individual symbiont-housing subcompartments there appears to be continuity between the lumen of a large majority of these subcompartments (Fig. 2h, i and Fig. S4, S5) allowing metabolite exchange between symbionts and with the external environment. The WGA staining experiment indicates that this exchange is rapid and extensive.

The specificity of symbiosomes for colonizing prokaryotes can be multifactorial, influenced by host and symbiont metabolic complementarity, receptor recognition, and host secondary metabolite production, which inhibits disruptive colonizers or pathogens[39]. Intriguingly, the *Anaeramoeba* genomes sequenced herein have many LGT-derived genes previously implicated in protein-protein interactions and cell-cell adhesion in host-symbiont systems[18], as well as several non-ribosomal peptide synthases (Supplementary Data 7) whose products might be influencing which bacteria can successfully colonize the symbiosome.

Hydrogen depletion is likely to be an important reason for host dependence on the *Anaeramoeba* symbionts. However, the large repertoire of host-encoded vitB12-dependent enzymes includes essential proteins involved in hydrogenosome metabolism and nucleotide scavenging, suggesting that a degree of metabolic complementarity exists and serves as a stabilizing factor. The lateral acquisition in *A. ignava* BMAN of *cobC*, encoding the final enzyme in

vitB12 biosynthesis, in *A. ignava* BMAN may help to optimize the synthesis of vitB12. Diatoms have been shown to strongly induce vitB12-binding proteins under limiting growth conditions[40] and modeling predicts active vitB12 release from bacteria in co-cultures[22]. How vitB12 is harvested from bacteria by microbial eukaryotes is not clear but directed release and lysis have both been suggested[22,41].

Whereas evidence for the obligate nature of the *A. flamelloides* symbionts is strong (i.e., the high degree of genome degeneration), the degree of autonomy of the *A. ignava* symbiont is less clear. Although all three *Anaeramoeba* strains examined here have Desulfobacteraceae symbionts, we do not know whether all Anaeramoebae exclusively establish partnerships with this group of bacteria or whether they can harbor other compatible strains or consortia that would similarly deplete hydrogen and provide vitB12. Hydrogen consumption may be carried out by other taxa such as *Arcobacter* spp[42]., some of which are predicted to also be vitB12 prototrophs, or additional Deltaproteobacteria that are present in our *Anaeramoeba* cultures (Fig. S7). Our LGT analyses indicate that no single bacterial lineage has disproportionately contributed to the gene repertoire of *Anaeramoeba*, which is reminiscent of other symbiotic systems in protists[43] and insects[44]. Our data and these other examples are consistent with the shopping bag model of gene acquisitions in relation to the evolution of symbiont-derived organelles[45]. Although we detected several cases of transfers from Desulfobacteraceae, they are not obviously from the resident symbiont lineages. Some of the LGTs we observed might also be derived from past symbionts that have since been replaced by the current lineages.

The numbers of predicted foreign genes in *Anaeramoeba* is higher than for most microbial eukaryote genomes investigated[17]. Many of these LGTs appear to be related to adaptation to life in low-oxygen environments, and only a few show obvious links to the establishment and maintenance of the symbiosis with their Desulfobacteraceae symbionts. The fact that the Desulfobacteraceae are not found to be major LGT donors suggests that the phagocytic capacity of *Anaeramoeba* may give rise to more foreign genes than does the symbiosis itself; symbiosome-associated bacteria may only rarely be fully internalized and digested, thus providing few opportunities for gene transfer.

We have provided deep insight into the evolution of a unique, elaborate symbiosome in an anaerobic protist lineage. We have revealed the detailed structure of this organelle as well as some of the metabolic interactions and selective pressures that led to its establishment, and the evolutionary consequences on the genomes of the partner organisms. Future studies will focus on dissecting the fine-scale structure and function of the *Anaeramoeba* symbiosomes and host-symbiont interactions.

## Methods

### Culturing and harvesting of *Anaeramoeba*

*A. flamelloides* BUSSELTON2 and SCHOONER1 and *A. ignava* BMAN cells were propagated as described in ref. 7. Large-scale cultures of *Anaeramoeba* were grown in 0.25× ATCC medium: 1525 Seawater 802 (SW802) medium media in fully filled 550 mL Cellstar® T175 low-profile flasks (Grenier Bio-One GmbH). Cells were dislodged with a cell scraper and split 1:1 with fresh media every 2–4 days for *A. flamelloides* strains or 7–10 days for *A. ignava* BMAN.

Large-scale cultures were harvested by decanting the culture supernatant and rinsing the ameba monolayer in each flask with 250 mL 1× Artificial Seawater (ASW) (per liter medium: 24.72 g NaCl, 0.67 g KCl, 1.364g $CaCl_2 \times 2H_2O$, 4.66g $MgCl_2 \times 6H_2O$, 6.29g $MgSO_4 \times 7H_2O$, 0.18 g $NaHCO_3$-). Cells were detached by cold-shock by adding 35 mL of ice-cold 1× ASW and immersing the flasks in ice-slush for 15 min. Percussive force was used to ensure efficient cell detachment. The cell suspensions were centrifuged at $500 \times g$ for 8 min at 4 °C and the combined cell pellets were resuspended in 10 mL ice-cold

1× ASW. Centrifugation was repeated as above, and the cell pellet was processed further for RNA or DNA extraction. Cultures of *Anaeramoeba* strains are available upon request.

## RNA extraction and sequencing

*A. flamelloides* BUSSELTON2 and *A. ignava* BMAN cells were harvested as above and total RNA was extracted using TRIzol™ Reagent (Thermo Fisher Scientific) according to the manufacturer's recommendations. Ten µg of total RNA was treated by the TURBO DNA-free™ Kit (Thermo Fisher Scientific) then treated with the DNase inactivation reagent. Total RNA was submitted to Génome Québec for sequencing. Libraries were made using the Illumina TruSeq RNA strand-specific sequencing kit and were sequenced on an Illumina HiSeq 4000 using 100 bp paired reads. Illumina reads were quality checked using FastQC v.0.11.5 (http://www.bioinformatics.babraham.ac.uk/projects/fastqc) and trimmed using Trimmomatic v0.36[46].

## Enrichment of prokaryotic mRNA and sequencing

Total RNA from enriched *A. ignava* BMAN and *A. flamelloides* BUSSELTON2 samples were treated by Terminator Exonuclease (Epicentre) according to the manufacturer's instructions. The total RNA samples were submitted to Génome Québec where polyA+ selection was performed to remove eukaryote mRNAs. One µl of supernatant from these RNA selections was used as input into the fragmentation step ahead of first-strand cDNA synthesis in the TruSeq protocol. The final library was sequenced on the Illumina NovaSeq 6000 S2 using 150 bp paired reads. Illumina reads were quality-checked and trimmed as above.

## DNA extraction and short-read sequencing

DNA was purified from large-scale cultures using the MagAttract HMW gDNA kit (Qiagen) using the tissue lysis protocol, then further purified on the GenomicTip G/20 column (Qiagen) by the manufacturer's protocol. Sample quality and quantity were assessed by agarose gel electrophoresis, UV specrophotometry and the Qubit™ dsDNA BR Assay Kit (Thermo Fisher Scientific).

DNA samples for Illumina short-read sequencing were submitted to Génome Québec for the construction of shotgun and PCR-free shotgun libraries (see Supplementary Data 8) using the Illumina TruSeq LT kit. The libraries were sequenced on an Illumina HiSeq X using 150 bp paired reads. Reads were quality-checked and trimmed as above.

## Long-read sequencing and basecalling

Genomic DNAs (1–5 µg) prepared as above were processed using Oxford Nanopore LSK108, LSK109, or LSK308 kits to construct sequencing libraries. The libraries were loaded on FLO-MIN106 (R9.4 or R9.4.1 pore) or FLO-MIN107 (R9.5 pore) flow cells and sequenced on the MinION Mk2 nanopore sequencer (Oxford Nanopore) running the MinKNOW control software. The fast5 files were basecalled to fastq format using Guppy v2.3.5 (Oxford Nanopore). The fastq reads were trimmed using Porechop v0.2.3_seqan2.1.1 (https://github.com/rrwick/Porechop) with the `--discard_middle` flag.

## Assembly and correction

The *A. ignava* BMAN read set was assembled using ABruijn (v1.0)[47] using default parameters. *A. flamelloides* BUSSELTON2 and SCHOONER1 read sets were assembled in metagenomics mode (--meta) using Flye (v2.4) with 3000 bp min overlap and Flye (v2.4.2)[48] with 1500 bp min overlap respectively.

Read mapping steps for Illumina short-reads were performed by bowtie2 (v2.3.1)[49] and Nanopore reads were mapped by minimap2 (v2.10-r761)[50] with parameters (`-ax map-ont`).

The BMAN assembly was corrected by three rounds of Racon (v0.5)[51] followed by Nanopolish v0.8.4 (nanopolish-git-dec-18-2017)[52].

The final BMAN assembly was obtained by two rounds of Pilon (v1.22) polishing employing (--mindepth 5 --fix bases) parameters[53].

The BUSSELTON2 and SCHOONER1 assemblies were corrected by four rounds of Racon (v1.4.13) with settings: `-u -m 8 -x -6 -g -8 -w 500`. Draft genomes were polished using Medaka v0.6.2 (https://github.com/nanoporetech/medaka) using the (`-m r941_trans`) model. The final assemblies were generated by five rounds of Pilon (v1.23)[53] polishing employing (--mindepth 1 --fix bases,amb) parameters.

## Genome classification and binning

Genome binning was performed manually utilizing the combined evidence of mapped polyA+ selected RNA sequencing data, sequence similarity searches (blastn and blastx) against the NCBI nr database, long-read coverage information, and GC-content of contigs. Chimeric and/or misassembles were identified by consistency of long-read mapping and split manually at the read-mapping border to retain the eukaryotic part of the contig. The presence of spliceosomal introns with GT-AG boundaries in genes was the main criterion for assigning a contig as being eukaryotic.

RNAseq data for *A. ignava* BMAN and *A. flamelloides* BUSSELTON2 were mapped using Hisat2 v2.1.0[54] using (`--rna-strandness RF --phred33 --max-intronlen 10000 -k 2`) flags. For *A. flamelloides* SCHOONER1, the BUSSELTON2 dataset was mapped with Hisat2 v2.1.0[54] using relaxed parameters (`--phred33 --max-intronlen 10000 -k 2 --mp 1,1 --sp 20`).

The GC% for each contig was calculated by `countgc.sh` in the BBMap package v38.20 (sourceforge.net/projects/bbmap/).

## Hybrid assembly of symbiont genomes

Long- and short sequence reads mapping to each Desulfobacteraceae genome were extracted and re-assembled using the hybrid-assembler Unicycler. For *A. flamelloides* SCHOONER1, the long-read data was assembled using Flye (v2.4.2 with --meta flag)[48]. A detailed description of symbiont genome assemblies can be found in Supplementary Note 4. ANI values were calculated using the OAT software[55].

## Prokaryotic annotation

The reassembled symbiont genomes were annotated by Rapid Annotation using Subsystem Technology v2.0 (RAST[56]:) (Sym_BMAN; 2294.7, Sym_BUSS2, 2294.12; Sym_SCH1 2294.13). Insertion sequences were predicted using the ISsaga v2.0 webserver (http://issaga.biotoul.fr/ISsaga2/). Pseudogene candidates were identified using Pseudofinder v0.11[57] and additionally refined by manual curation. Synteny was calculated using Sibelia v3.0.7[58]. Circos plots were created with Circa (http://omgenomics.com/circa).

## 16S rDNA amplicon typing

The *Anaeramoeba*-enriched cell material was prepared by cold-shock and monolayer rinsing as described above. The supernatant samples were obtained by pelleting 10 mL of culture supernatant poured off prior to monolayer rinsing. The samples were extracted using the DNeasy PowerSoil Pro Kit (Qiagen) according to the manufacturer's instructions.

The V4V5 (Bacteria) or V6V8 (Archaea) regions of the 16S rRNA genes were amplified and sequenced at the Integrated Microbiome Resource (IMR) at Dalhousie University, as described in ref. 59. Bioinformatic processing of raw reads was carried out by IMR (see ref. 59), updated in the Standard Operating Procedures (SOPs) for creation of ASV in QIIME2 (v2019.7) as outlined on the current MicrobiomeHelper website (https://github.com/LangilleLab/microbiome_helper/wiki).

## Phylogenomics

Phylogenomic analysis was performed on the symbiont genomes and genomes from Desulfobacterales (NCBI:txid213118) supplemented by

data from outgroup taxa. The initial selection of representative genomes and marker genes/proteins was done using phyloSkeleton v1.1[60] from 353 Desulfobacterales genomes from NCBI. One representative genome per genus was selected in Desulfobacteraceae except for *Desulfobacter* where all available genomes were selected. The Bac109 protein markers were identified with phyloSkeleton v1.1[60] and only genomes with >80% of the marker genes were selected for further analyses. The UPF0081 marker was removed since it was only present in the out-group genome but not in any of the in-group genomes. Each of the 108 remaining marker proteins was aligned by MAFFT-linsi v7.458[61] and trimmed by BMGE v1.12[62], using the BLOSUM30 matrix and stationary-based character trimming. Alignments were concatenated and a phylogenetic tree was inferred by IQTree v2.0.3[63] using the LG4X[64] substitution matrix with 1000 ultrafast bootstraps[65]. The resulting tree was inspected, closely related taxa were removed, and the procedure was repeated. The final dataset consisted of 35 genome-derived sequences: *Desulfovibrio desulfuricans* ND132 (1 genome), Desulfobacteraceae (23 genomes), *Desulfobacter* (8 genomes) and the *Anaeramoeba* symbionts (3 genomes). A maximum-likelihood tree was inferred using IQ-TREE v2.0.3[63] using the LG + C60 + F+ Gamma mixture model[66]. The PMSF model[67] generated from the above guide-tree and fitted model was used to perform 100 nonparametric bootstrap replicates.

### Protein-coding gene family expansions and contractions

We employed the PANTHER family database (v15)[68] to examine which protein families may have undergone expansions or contractions. Assignment of proteins to families was done with direct hidden Markov models (HMM) searches and the Panther Score tool[69] (set for hmmsearch with HMMER v3.1b2[70]) using the PANTHER family database with e-value cut-off <1e-5. Once classified, proteins assigned to each PANTHER family were counted to investigate relative family expansion or contraction in any *Anaeramoeba* species relative to other groups. (Protein families that were taxon-specific, exclusive to each group being compared, apparently heavily expanded in diplomonads, *Trichomonas, Dictyostelium* or *Naegleria*, and those with median counts of zero for any compared group were not considered in the expansion/contraction analysis.) To detect signatures of expansion/contraction, outlier values within each group were replaced by the median value of their respective group when the value had a $1.5 > z$-score $< -1.5$. Data were normalized by the median value of their respective families and $\log_2 FC$ among groups was calculated. To identify what metabolic pathways were relevant for further analyses, we investigated families with signature of expansions or contractions $\geq 5$-fold ($2.35 > \log_2 FC$ or $\log_2 FC < -2.35$). Relative contractions and expansions are reported for *Anaeramoeba* taxa and protein families were considered "core' if they were present in all studied proteomes and "accessory' if they were missing in at least one. Possible expansions/contractions were visualized with Circos[71]. A large proportion of the protein families expanded beyond $\geq 5$-fold belong to RNA synthesis, DNA synthesis and repair, and membrane trafficking metabolisms. Hence, we carried out pathway reconstruction for RNA synthesis and membrane trafficking (see below) by validating protein orthology and recording absence/presence patterns. In the case of RNA systems, we first classified query proteins of interest (from yeast or human) with PANTHER Score (as above) to identify their PANTHER family. Query proteins and proteins identified for each PANTHER family of interest were then aligned together with a random taxonomic sample from the same family of interest from the PANTHER database. Alignments were trimmed to carry out phylogenetic reconstructions as described in ref. 72.

For membrane-trafficking proteins, *Anaeramoeba* datasets were searched by BLAST using queries from previous studies[73–75]. Identified Rab and TBC proteins were subjected to orthologous clustering by OrthoFinder v2.0.0[27] including proteins from human, cnidarian *Nematostella vectensis*, and the heterolobosean *Naegleria gruberi*.

Phylogenetic analyses for selected Rab sub-families and TBC proteins were conducted. Sequences were aligned by MAFFT v7.458[61] under L-INS-i strategy, and poorly aligned positions were removed by trimAl v1.4.rev15[76] using -gt 0.8. Maximum-likelihood trees were inferred by IQ-TREE v1.6.8[77] using the PMSF method[67] and the LG + C20 + F + G model, with the guide tree inferred under the LG + F + G model. Ultrafast bootstrap (UFBOOT2) branch supports were obtained with 1000 replicates.

### Lateral gene transfer in *Anaeramoeba*

In order to assess LGT from prokaryotic and viral donors, we conducted a large-scale screen of the predicted proteomes of the three *Anaeramoeba* genomes. Initial clustering was performed using Orthofinder v2.5.4[27] with the following outgroup proteomes: *Arabidopsis thaliana, Dictyostelium discoideum, Acanthamoeba castellanii* str. Neff, *Giardia intestinalis, Homo sapiens, Kipferlia bialata, Monocercomonoides* sp., *Naegleria gruberi, Trepomonas* PC1, *Trypanosoma brucei, Trichomonas vaginalis, Saccharomyces cerevisiae*. In some instances, Orthofinder appeared to cluster proteins that were assigned different PANTHER annotations obtained in the protein-coding gene family expansion/contraction analysis, indicating they were unlikely to be true orthologs. In such cases, clusters were further split according to the PANTHER annotations.

Proteins for phylogenetic trees were inferred by gathering homologous sequences in the nr database using blastp using an E-value cutoff of $1 \times 10^{-5}$. All *Anaeramoeba* orthologs/paralogs constituting a cluster together with the 500 best database hits to each were grouped. Clusters containing at least one prokaryotic or viral protein together with *Anaeramoeba* orthologs/paralogs were aligned using MAFFT v7.310 [61] with the default setting and sites were selected using BMGE v1.0[62]. Initial phylogenetic reconstructions were done using FastTree v1.0.1[78]. These phylogenies were used to reduce the taxonomic redundancy of the initial sequence files using in-house scripts. The reduced files were realigned using MAFFT v7.310 [61] with the accurate option (L-INS-i), and sites were selected using BMGE v1.0[62]. Phylogenetic reconstructions were done using IQ-TREE multicore v1.5.5[77] with the LG4X model and 1000 UFBOOT for alignments $\geq 80$ sites (shorter alignments were treated separately, see below). Phylogenetic trees were then parsed to find putative LGTs, using the following criteria:

i) When *Anaeramoeba* sequence(s) and prokaryotic or viral sequences constituted a clade supported by an UFBOOT value of $\geq 70\%$, the *Anaeramoeba* homologs were considered as possible LGTs. To allow for mis-annotation, one other sequence in that clade could be eukaryotic.

ii) When no clades were well-supported, if $\geq 95\%$ of the sequences in the tree were prokaryotic and/or viral, then this was counted as an LGT.

iii) The taxonomic group of the donor could be inferred when >50% of the taxa in a clade containing the *Anaeramoeba* protein(s) were from a particular taxonomic group.

iv) Trees were estimated for only a subset of OGs that contained at least one prokaryotic or viral taxa and >80 sites. However, when all sequences constituting the cluster, besides *Anaeramoeba*, were prokaryotic or viral, it was counted as LGT.

### FISH probe design and testing

The symbiont 16S rDNA sequences were extracted from the RAST annotations and aligned by MAFFT v7 (https://mafft.cbrc.jp/alignment/software/) to selected full-length 16S rDNA sequences selected from the Ribosomal Database Project (RDP).

Probes targeting Sym_BUSS2 and Sym_SCH1 to the exclusion of outgroups in the 16S rDNA alignment were designed using Decipher[79]. The probes were evaluated by matching against the SILVA and RDP databases and did not match any other sequences in the database at 1

mismatch. The probes were ordered from biomers.net GmBH. Probe sequences, fluorophores, and hybridization conditions can be found in Supplementary Data 9. Control probing for probe BUSS/SCH is shown in Fig. S7.

### Fluorescence in situ hybridization (FISH)

*Anaeramoeba* cell cultures in 10 mL slanted culture tubes were decanted and the adhered cells were resuspended in the remaining volume of culture media (typically 200 µl) by percussive shock. The suspended cells were applied to Teflon multi-well microscope slides (Electron Microscopy Sciences (ER-264)) for 5 min at room temperature. The cells were fixed for 10 min at room temperature by adding a suitable volume of 16% methanol-free formaldehyde (w/v) (Pierce-Thermo Fisher Scientific, Cat No 28908) to give a final concentration of 4% paraformaldehyde in ASW. The cells were rinsed in $2 \times 50$ mL distilled water for a total of 3 min and then air-dried. The slide was immersed in 100% ethanol for 5 min and air-dried again. Hybridization buffer (20 mM Tris-HCl, pH 7.6, 0.01% SDS, 900 mM NaCl) with an appropriate concentration of formamide and probe (5 ng/µl) (Supplementary Data 9) was added to dried cells and incubated for 2–3 h at 47 °C in a moisture chamber. Post-hybridization, the slides were rinsed with buffer (20 mM Tris-HCl, pH 7.6, 0.01% SDS, 5 mM EDTA and an appropriate NaCl concentration depending on the %FA in the hybridization buffer; see Supplementary Data 9) and incubated with 50 mL for 45 min at 48 °C, finally rinsed in water for 40 s and air-dried. The slides were mounted in ProLong™ Diamond Antifade Mountant with DAPI (Thermo Fisher Scientific, Cat No P36971) using #1.5H cover slips (Ibidi, Cat No 10811). The slides were incubated at room temperature ≥ 12 hours before imaging.

Images were acquired using either wide-field microscopy on a Zeiss Axio Imager Z2 or by confocal microscopy on a Zeiss LSM 710 or Leica SP8. Pseudo color, merging of channels and projections were made in Zeiss ZEN v3.1 or BioImageXD[80].

### Scanning electron microscopy

*A. flamelloides* BUSSELTON2 cells were cultured and harvested as described above with cells finally being resuspended in 500 µL of ice-cold filtered growth media. 50 µL of cells were transferred onto poly-L-lysine-coated 12 mm round coverslips, left to adhere for 5 min at room temperature and immediately fixed with a drop of 25% glutaraldehyde and $OsO_4$ vapor for 1 h. After fixation, the coverslips were washed three times in filtered ASW, and then subjected to a dehydration series of ethanol-water mixtures, as follows: 30%, 50%, 70%, 80%, 90%, 95%, 100% (three times). This was followed by critical-point drying with $CO_2$ on a Leica EM CPD300, then an ~15 nm Au-Pd coat was added with a Leica EM ACE200 sputter-coater. Samples were imaged on a Hitachi S4700 scanning electron microscope.

### Transmission electron microscopy

Fixation (both chemical and cryofixation), embedding, sectioning, and TEM were conducted as described in detail by ref. 7.

### Focused-ion-beam scanning electron microscopy (FIB-SEM)

Cells were adhered to the surface of a gridded MatTek dish and fixed with 2.5% glutaraldehyde (TAAB) in 0.1 M PHEM-buffer. All samples were processed using a Pelco Biowave Pro+ microwave tissue processor (Ted Pella, Redding, CA) according to[81] with minor modifications: no calcium was used during fixation and the contrasting steps with lead aspartate was omitted to reduce the risk of overstaining. Samples were detached from the glass using liquid nitrogen and glued to an SEM-stub with epoxy and silver glue. Samples were further coated with 5 nm platinum to reduce charging. Volumes were acquired using a Scios dual-beam (Thermo Fischer Scientific) with the electron beam operating at 2 kV/0.2 nA detected with a T1 In-lens detector. To automate volume acquisition, we used the Auto Slice and View

4 software provided with the microscope. A 700 nm protective layer of platinum was deposited on the selected area before milling. A FIB-SEM volume of 1780 slices of *A. flamelloides* BUSSELTON2 was acquired close to isotropic resolution ($6.7 \times 6.7 \times 7$ nm). Volumes were further registered and processed using the ImageJ plugins Linear alignment by SIFT and Multistackreg. After registration the volumes were converted to mrc-files and header was modified to recover the pixel-sizes that got lost during conversion.

### Segmentation and visualization of cell structures

We segmented eight cell structures (nucleus, symbionts, hydrogenosomes, symbiosome-membrane, dense granules, other prokaryotes, plasma membrane, and the acentriolar centrosome with individual microtubules) using Microscopy Image Browser v2.84[82,83]. The nucleus was segmented using the Graphcut semi-automatic segmentation function in MIB. Symbionts, symbiosome-membrane, and hydrogenosomes were segmented by the deep-learning segmentation tool in MIB (DeepMIB). Briefly, symbionts and hydrogenosomes were manually annotated in a 50-slice segment of the FIB-SEM volume abundant in symbionts and hydrogenosomes. Image segments (patch size $256 \times 256$) were extracted and used to train DeepLabV3 ResNet50 model using default settings. The trained model was then used to predict segmentations for symbionts and hydrogenosomes. The resulting symbiont model was manually refined using the MIB segmentation tools. The hydrogenosomes and dense granules were predicted together by the classifier and were manually separated by hand using the MIB segmentation tools. Other prokaryotes were partially annotated by the symbiont+hydrogenosome model and were separated manually for additional curation by hand using the MIB segmentation tools. The symbiosome-membranes were predicted by training a model that segments whole symbionts (symbiont+symbiont subcompartment and membrane) as described as for symbiont and hydrogenosomes above. The outer membranes were obtained by eroding the whole symbiont predictions by 3 pixels (approximated thickness of the symbiosome-membrane) and using this selection as a mask to cut out the outermost 3 pixels of the whole symbiont model. The plasma membrane was manually annotated using black-white thresholding of brush-traced selections every 5–10 slices and shape interpolation was applied between those slices. In segments where the membrane was sharply shifting between slices individual slices were segmented by hand using black-white thresholding of brush selections. The acentriolar centrosome was segmented using Graphcut and individual microtubules were manually annotated using the brush tool. Symbiosome subcompartment connectivity was manually traced, annotated and visualized in MIB v2.84[82,83]. The volumes were rendered using ORS Dragonfly v2022.2.0.1399. Further information about FIB-SEM is reported in Supplementary Note 5.

### Tubulin staining

For tubulin staining cells were harvested and fixed for 10 min in 4% paraformaldehyde according to the procedure described in the "Fluorescence in situ hybridization (FISH)" section. Fixative was removed, and the cells were washed twice with ASW and once with PBS. Cells were then incubated 15 min in PBS with 50 mM $NH_4Cl$ to quench remaining aldehyde fixative. The cells were washed twice with PBS and permeabilized for 15 min in PBS with 0.1% Triton X100. Cells were blocked in antibody dilution buffer (ADB - 1% BSA-c (Aurion) in PBS with 0.1%Triton X100) for 1 h at room temperature. The cells were then incubated with primary antibodies diluted in ADB (TAT1; 1:200 dilution or KMX-1; 1:200 dilution) overnight at 4 °C in a moisture chamber. The slides were droplet-washed six times using excess ADB and incubated 1 hour at room temperature with secondary antibody (goat anti-mouse Alexa Fluor 594, 1:250 dilution, Thermo Fisher Scientific, Cat No A-11032). The cells were washed six times using ADB, twice with PBS, and then mounted in ProLong™ Diamond Antifade

Mountant with DAPI (Thermo Fisher Scientific, Cat No P36971) using #1.5H cover slips (Ibidi, Cat No 10811).

Images were acquired using wide-field microscopy on the Zeiss Axio Imager Z2 or by confocal microscopy on Zeiss LSM 710 or Leica SP8. Pseudo color, merging of channels and projections were made in Zeiss ZEN or BioImageXD[80].

## Wheat Germ Agglutinin (WGA) staining

For WGA staining two different protocols were used, the first was a live stain protocol, and the second relied on staining after aldehyde fixation. In both protocols, cells were harvested and attached to multi-well slides according to the FISH procedure (above). Live stain cells were washed twice using 100 µl ASW and then incubated for 10 min in 50 µg/ml WGA-CF633 conjugate (Biotium, Cat No #29024-1) in ASW. The stain was removed, and cells were washed twice in 100 µl ASW and then post-fixed in 4% formaldehyde in ASW for 10 min at room temperature.

For the aldehyde fixation protocol, cells were fixed in 4% formaldehyde in ASW immediately after attachment and the first two washes. The fixative was removed, and the cells were washed twice in 100 µl ASW.

Each slide was rinsed twice in a large volume of distilled water for 1 min 30 s each and air-dried. The slides were incubated 5 min in 100% ethanol, air-dried and the slide was mounted and imaged according to the procedure described in the FISH section.

## Reporting summary

Further information on research design is available in the Nature Portfolio Reporting Summary linked to this article.

## Data availability

Sequencing reads and the annotated genomes of *A. flamelloides* BUS-SELTON2, *A. flamelloides* SCHOONER1, *A. ignava* BMAN, Sym_BUSS2, Sym_SCH1 and Sym_BMAN were deposited to NCBI under the BioProject number PRJNA634776 [https://ncbi.nlm.nih.gov/bioproject/634776]. FIB-SEM raw data, aligned mrc-files and segmentation models in ORSObject format are available from Figshare: https://doi.org/10.6084/m9.figshare.24033777. Tree-files and alignments are available from Figshare: Fig. 3 at https://doi.org/10.6084/m9.figshare.20375619, Figure S10 at https://doi.org/10.6084/m9.figshare.20375601, Figure S19 at https://doi.org/10.6084/m9.figshare.22193497. Animations are available at Figshare: Movie S1 at https://doi.org/10.6084/m9.figshare.27108724, Movie S2 at https://doi.org/10.6084/m9.figshare.27108751.

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

## Acknowledgements

Gordon Lax and Yana Eglit are acknowledged for help during SEM sample preparation and imaging. The majority of the work and J.J.H. were supported by a Foundation grant (FRN-142349) from the Canadian Institutes of Health Research awarded to A.J.R. J.J.H. was additionally supported by a grant from Vetenskapsrådet, (VR-NT grant 2022-04490). The research was also supported by grant GMBF12188 from the Gordon and Betty Moore Foundation. Archibald Lab contributions to this study were supported by the Gordon and Betty Moore Foundation (GBMF5782) and a Discovery Grant from the Natural Sciences and Engineering Research Council of Canada (RGPIN-2019-05058). Work from the Čepička lab was funded by Czech Science Foundation grant no. 21-30563S. Computational resources were provided by the e-INFRA CZ project (ID:90254) supported by the Ministry of Education, Youth and Sports of the Czechia. Research in the Dacks Lab was supported by grants from the Natural Sciences and Engineering Research Council of Canada (RES0043758, and RES0046091). The authors acknowledge the facilities and technical assistance of the Umeå Core Facility Electron Microscopy (UCEM) at the Chemical Biological Centre (KBC), Umeå University, a part of the National Microscopy Infrastructure NMI (VR-RFI 2019-00217).

## Author contributions

Conceptualization, J.J.-H., and A.J.R.; Investigation, J.J.-H., I.Č.; Formal analysis, J.J-H, L.G.L., D.E.S.L., B.A.C., K.Z., C.W.S., S.P.; Resources, I.Č., A.J.R.; Writing—Original Draft, J.J.-H., L.G.L., D.E.S.L., B.A.C., K.Z., J.B.D., J.M.A. and A.J.R.; Writing – Review & Editing, J.J.-H, L.G.L., D.E.S.L., B.A.C., K.Z., I.Č., C.W.S., S.P., J.B.D., J.M.A. and A.J.R.; Funding Acquisition, J.J.-H., I.Č., J.B.D., J.M.A., and A.J.R.

## Funding

## Competing interests

The authors declare no competing interests.
