## [Peer Review file · Nature Communications]

A unique symbiosome in an anaerobic single-celled eukaryote

Corresponding Author: Dr Jon Jerlström-Hultqvist

Version 0:

Reviewer comments:

Reviewer #1

(Remarks to the Author)

Jerlstrom-Hultqvist et al. dig deeply into the symbiosis between two *Anaeramoeba* species and their Desulfobacteraceae symbionts. Endosymbionts from this sulfate-reducing bacterial group are known in various anaerobic protists, and some genomic analyses have hinted at the relationships/ interdependencies between protists and sulfate-reducing symbionts. EM has also provided prior evidence for a close physical association between endosymbionts and hydrogenosomes/MRO's in various protist taxa. However, the unique contribution of this study is a beautiful synthesis of genomics (both whole genome sequencing and transcriptomics) of host and symbionts, FISH, and 3D reconstruction using SEM to describe the underpinnings of the symbiosis for these holobionts. The authors show that the symbionts reside within a membrane network associated with cytoplasmic hydrogenosomes and that this network is connected to the cell exterior, facilitating ongoing sulfate transfer to the symbionts. As such it fills an important gap in our understanding of the possible trajectory toward endosymbiosis whereby in cases like this the symbionts may be transitionally "caught" between being ectosymbionts and fully enclosed endosymbionts. Of particular interest was genomic evidence (e.g. IS elements, flagellar loss and pseudogenization) for divergence of these common sulfate reducing bacterial types from free-living representatives. Host-symbiont metabolic connectivity and interaction is supported by presence of type VI secretion systems in the symbionts, genome expansions (e.g., for ribosomal proteins and membrane trafficking), vitB12 dependence of host on symbionts, membrane complexes, genome erosion buffering mechanisms (e.g., high expression of GroEL/S), etc. The evolutionary history of these symbioses are explored deeply, as are evidences for LGT from diverse sources over the history of these eukaryotic hosts. The supplementary files are a wealth of data and analyses (extensive phylogenies and genome analyses) to be explored in numerous ways by interested researchers who study symbiosis. It is a shame that some real gems are relegated to the supplementary files. The videos are wonderful. This paper will be of interest to the readers of Nature Communications and will be well-cited as a model study of symbiosis. The methodologies are sound, and work meets the expected standards in the field. Enough detail for methods is provided (see notes in comments below), and the work supports the conclusions. Some minor comments for the authors to consider:

1. Abstract line 29: perhaps "highly specialized structures of symbiotic origin..."? To differentiate between the organelles of symbiotic origin and the current symbioses?
2. Abstract line 47: "enhanced" instead of "enriched"?
3. Line 58 fix grammar
4. Line 72: Reference 4 is ok here, but this paper is mainly the genomics investigation Beinart made into this symbiosis, and the original report of the symbiosis containing more microscopy showing what you highlight in this sentence is the paper Edgcomb et al. 2011 *Frontiers in Microbial Physiology*.
5. Line 85: consider making the definition of 'symbiosome' clearer at first introduction of the term since it is not frequently used yet. A reference or brief definition, or move line 91 up to help clarify.
6. Line 101 and Figure 1B-E it would help to more clearly mark the vacuoles you are referring to for the reader in the figure. The arrows and boxes are very hard to see. Color maybe?
7. Figure 2 legend for h-i: Please separate notations for each panel and check the color notations. It is confusing to have plasma membrane in red and then in yellow next sentence. I don't see red plasma membrane. Also clarify "Two symbionts whose membrane are contiguous with the plasma membrane are shown in red." Should this be blue? Last sentence of legend needs clarification too.
8. Figure 3. Suggestion that it would be nice to add negative control images for each probe in the supplementary since the area of the nucleus often shows signal for the bacterial probes. This can be due to orientation. The FISH protocol does not

mention use of positive and negative controls nor Table S12.

9. Line 145: is it possible to say based on microscopy whether these estimates for numbers of symbionts per host are supported?

10. Line 218: You discuss the cob genes for B12 synthesis lower in the manuscript, present in the symbiont, and you say that ROS defense mechanisms exists in the symbionts, and so it would be also interesting if you looked for evidence of the anaerobic B12 synthesis pathway in the symbionts. The specific genes involved are *cbi*.

11. Lines 226-230: Can you comment here or in discussion on the significance of such high A+T?

12. Line 234: Section on B12 dependency. Perhaps point out that in some cases, B12 can also provide methyl groups (methylcorinoids) that support C1 metabolism? See Author summary in Price et al. 2021.

13. Line 347 on: where you describe Rabs and their function, please acknowledge their possible role in signal transduction and as molecular switches that regulate formation and transportation of vesicles (e.g., Hutagalung and Novick 2011).

14. Any evidence for archaeal symbionts in Metamonada? Thinking of your LGT data.

15. Lines 408-410: They maintain connections to the outside environment presumably due to metabolic constraints of the symbionts that require sulfate, however is it possible also that this is also maintained by modifications/restrictions in the membrane-trafficking system of the host that prevent the vacuoles from fully forming and hence cutting them off from direct contact with the outside environment? Your results section suggests this might be the case, so perhaps mention here? I apologize if in the extensive documents I missed this!

16. Lines 514-7: Please clarify for the reader why different assemblers and min. overlaps were used for the three host genomes.

17. Table 1. Can you perhaps clarify what exactly the column labeled "trees" is showing either by changing the column header or in the legend?

18. I may have missed it, but either in the reporting summary or text documents, please be sure that readers know how /where to acquire cultures of these organisms if desired.

19. Any reason why Delta495a probe images are not available in Fig. S6 for *A. flammelloides* BUSSELTON2 or SCHOONER?

20. Supplementary discussion of selenoproteins: Perhaps point out that use of selenoproteins and selenocysteins while present in certain anaerobic taxa is overall not a common trait, and their synthesis may be a trait acquired through horizontal transfer events (Peng et al. 2016).

Reviewer #2

(Remarks to the Author)

In this manuscript by Jerlström-Hultqvist and colleagues, the authors examined the symbiosis between multiple *Anaeramoeba* species and their Desulfobacteraceae symbionts. The manuscript is a methodological tour de force using most methods currently available for studying symbioses of protists. The authors generated not only complete genomes of several symbionts but also transcriptomes, high-quality host genomes, and multiple types of imaging data. The bioinformatics analyses are solid, and address a broad range of questions, including LGT and gene family expansions in the nuclear genomes. I do not find it necessary to comment on the genomics part of the manuscript because it's already very polished and on par with the highest standards in the field. Overall, my evaluation of the manuscript is very positive. My only suggestions for improvement concern the microscopy results, the appropriateness of the term symbiosome, and the manuscript not being accessible to a broad audience.

(1) I find the manuscript overwhelming and not very accessible to a broad audience. The authors did a great job of running all possible analyses. However, they included all the results in the MS which means that there's so much data that many interesting novel findings are buried in supplementary files (15 supplementary tables and 19 supplementary figures). The amount of data included in this paper, in my view, goes over the limits of what should be included in a single research article (it reads more like a monograph on everything we know about this particular symbiosis) which inevitably means that some results are only touched superficially, and a broad synthesis is missing.

(2) The vacuole/organelle housing all the symbionts is presented as the main finding of the manuscript. Like the term 'bacteriome' that is used broadly in multicellular eukaryotes for diverse organs that are composed of cells (bacteriocytes) housing endosymbionts, the term symbiosome is used here for a novel membranous organelle harboring symbionts. I fully agree that this finding is potentially extremely interesting. However, extraordinary claims require extraordinary evidence. The result is only supported by limited imaging data and is not properly discussed in the context of what exactly the organelle is. First, I am not 100% convinced that the single FIB-SEM dataset provides sufficient evidence that a single vacuole houses all the symbionts. Could you please expand the methods section with details on how exactly were the data segmented and what is the pixel size? If the resolution was sufficient for confidently segmenting membranes, why is the endomembrane system not shown? Does the 'symbiosome' somehow interact with (or is a part of) the ER and Golgi? Are any other compartments (such as food vacuoles) of *Anaeramoeba* also partly connected to the extracellular environment? What happens with the symbiosome after symbiont loss/replacement?

Expanding the symbiosome definition to apply to the *Anaeramoeba* system is, in my view, potentially misleading. The term symbiosomal membrane is used in many different systems (including protists) for a membrane of host origin surrounding a single symbiont cell (and symbiosome is used, relatively rarely, for the entire compartment). Please correct me if I'm wrong, but this seems to be the first time the term symbiosome is used for a host-derived vacuole network surrounding many symbiont cells. Since it has tubular connections leading to the cell surface, its origin and membrane composition are, however, less clear. Did it originate via a similar process that leads to what we usually call a symbiosomal membrane? The authors suggest that the organelle is reminiscent of phagosomes that have been delayed or frozen in their maturation

process (P14L347) but no experimental evidence supporting this hypothesis is included. In well-studied systems such as the Buchnera-pea aphid model, the symbiosomal membrane originates when extracellular symbionts are endocytosed by the posterior syncytial cytoplasm of blastula (Koga et al., PNAS 2012). Buchnera cells then divide into daughter cells that are both covered by the 3rd symbiosomal membrane. In the case of Anaramoeba, the symbiont cell division happens inside the vacuole which could suggest that this membranous organelle is different from what would be usually called a symbiosome. The authors should clarify what the organelle is and use the term consistently. Some of the descriptions from the text (see below) are too vague or even contradictory (vacuoles, P5L99).

P2L36: We show that the symbionts reside deep within a symbiosomal membrane network that is tightly associated with cytoplasmic hydrogenosomes, and, importantly, maintains connections to the Anaramoeba cell surface.

P4L79: These symbionts, although not in direct contact with the hydrogenosomes, are enveloped by an intricately organized membrane of host origin.

P4L91: ...the symbionts are situated within a symbiosome, which comprises an interconnected vacuolar network positioning the symbionts alongside host hydrogenosomes.

P5L99: ... a large mass of symbionts apparently housed in vacuoles tightly positioned close to, but not in direct contact with, hydrogenosomes ...

P6L128: ... a network of deep, anastomosing invaginations of the cell's membrane with extracellular opening.

P16L397: ...suggests that the membranous 'organ' housing the symbionts is a highly evolved structure that might allow them to selectively manage their captured symbionts and position them in tight association with their hydrogenosomes.

Reviewer #3

(Remarks to the Author)

This is an exciting and broadly interesting paper. I agree with the authors that not much is known about the cell biological structures of symbiotic systems, protists in particular, and that this paper provides a unique look into a very interesting protist-bacterial symbiosis. It is novel but also provides interesting comparisons with (comparatively better-studied) animal-microbe and plant-microbe symbioses. The paper has already changed the way I view symbiosis, by giving me a cool example of the cell biological Rube Goldberg machines host can make to enable their symbioses to work. This paper will be widely read, discussed, cited. It's exciting and high-quality work.

That being said, I do have one major issue, one minor (stylistic) issue, and several small points. I believe all of these points are addressable without further experiments, but might require making one or two more figures, as I explain below.

Major issue:

The critically important result that the bacteria exist in a connected network that has continuous exposure to the outside world is not clearly shown in the paper. The WGA result is very suggestive and I think a good experiment. The TEM in Figure 1f does show very close connections between hydrogenosomes and the host vacuolar membrane, but in this case the "narrow opening connected to outside media" could be a phagocytic event that is not yet complete. It *could* be a completed phagocytic event in which a connection to the outside world is retained, but it also might not be. One can see other bacteria that seem to be undergoing phagocytosis nearby (the bacterium at the lower-right corner of the dashed inset box, for example) and so in my mind this complicates this interpretation.

The 3D reconstructions of the FIB-SEM data in Figure 2 do not show this, either. Figure 2h does show connections to the outside world (the black arrows), and I agree that the shape of these connections are less likely to be captured in-process phagocytic events, but nowhere in Figure 2 are the networks of connections shown.

The closest the paper gets to showing the network is in Figure S4. Here, pairs of clearly interior bacteria are seen to have openings which might connect these bacteria to each other. But nowhere (that I can see) are connections traced from the surface to bacteria that are well inside the protist.

I strongly suspect that these data exist, and that the authors have seen these connections. (I was hoping to see this in the videos, but I didn't there, either.) The quality of the FIB-SEM and the TEM is good enough that I am certain this could be shown in the main paper. I would suggest that Figure 2 be re-worked. Some existing panels could be put into supplemental (maybe d,e,g, and i?) which would make space for a reconstruction that traces the plasma membrane, through a surface-adjacent symbiosome, to a neighboring symbiosome, to another, etc., such that the host-membrane connectivity between outside and interior symbionts is very clear. As the paper is now, a reader can somewhat piece this together, but the conclusion of complete interconnection now rests mostly on trust of the authors (of which I have a great deal, to be honest).

Minor issue:

The paper starts off strongly, with (mostly) experimentally and visually compelling data and results. The following sections on genomics are a bit of a tougher read, and seem about 50% too long. There is interesting stuff in there, and I think the main results should stay in the paper because they obviously contextualize the imaging, but many of the details could be moved to supplemental. Alternatively, maybe the genomics could be presented first, in a shorter format, to set up the question of how the symbiosis could possibly work when the symbionts need both tight connections to hydrogenosomes AND

connection to the outside world? People like riddles.

I can see it either way, and it's not my paper, but that's my honest assessment.

Small points:

1. The abstract mentions that the data supports the idea that the host "facilitates symbiont partitioning during cell division." I am not sure the data strongly support this point being included in the abstract, but perhaps I missed something.
2. What is the interpretation of Figure 1b? What are we to learn from it?
3. Figure legend for Fig 2h-I refers to red color but I think this is a mistake.
4. I had to look up anastomosing, which is fine, and I've learned something, but is there a simpler way to say this?
5. Lines 441-44: "If *Anaeramoeba* species are truly dependent on symbiont-provided vitB12, then this might also mean that cyclical symbiont-replacement, as seen in the ciliate *Euplotes* 41, is necessary to ensure that they receive a steady source of vitB12 in the face of symbiont genome erosion." Maybe, but it seems like if B12 is important genes for it would be under at least as strong purifying selection as all other genes important to the host, and so I don't know why you would expect turnover based on these genes in particular.
6. Line 445: Yes, this does support the shopping bag idea, but so also does the results from both other protist (10.1016/j.cub.2017.08.010) and insect (10.1016/j.cell.2013.05.040) symbioses. Would be good to contextualize these results a bit more with the literature.

Version 1:

Reviewer comments:

Reviewer #1

(Remarks to the Author)

I am satisfied with the responses to my questions and requests for revision. I understand that the paper is already quite dense, and can not delve further into new territory.

Reviewer #2

(Remarks to the Author)

The authors have sufficiently addressed most of my comments. I have no further questions.

Reviewer #3

(Remarks to the Author)

Sorry to be slow, was on vacation. All of my points have been addressed, thank you to the authors for that and congratulations on an outstanding contribution.

Below we have responded to the reviewers' comments and detail our revisions (in red).

REVIEWER COMMENTS

Reviewer #1 (Remarks to the Author):

Jerlstrom-Hultqvist et al. dig deeply into the symbiosis between two *Anaeramoeba* species and their Desulfobacteraceae symbionts. Endosymbionts from this sulfate-reducing bacterial group are known in various anaerobic protists, and some genomic analyses have hinted at the relationships/ interdependencies between protists and sulfate-reducing symbionts. EM has also provided prior evidence for a close physical association between endosymbionts and hydrogenosomes/MRO's in various protist taxa. However, the unique contribution of this study is a beautiful synthesis of genomics (both whole genome sequencing and transcriptomics) of host and symbionts, FISH, and 3D reconstruction using SEM to describe the underpinnings of the symbiosis for these holobionts. The authors show that the symbionts reside within a membrane network associated with cytoplasmic hydrogenosomes and that this network is connected to the cell exterior, facilitating ongoing sulfate transfer to the symbionts. As such it fills an important gap in our understanding of the possible trajectory toward endosymbiosis whereby in cases like this the symbionts may be transitionally "caught" between being ectosymbionts and fully enclosed endosymbionts. Of particular interest was genomic evidence (e.g. IS elements, flagellar loss and pseudogenization) for divergence of these common sulfate reducing bacterial types from free-living representatives. Host-symbiont metabolic connectivity and interaction is supported by presence of type VI secretion systems in the symbionts, genome expansions (e.g., for ribosomal proteins and membrane trafficking), vitB12 dependence of host on symbionts, membrane complexes, genome erosion buffering mechanisms (e.g., high expression of GroEL/S), etc. The evolutionary history of these symbioses are explored deeply, as are evidences for LGT from diverse sources over the history of these eukaryotic hosts. The supplementary files are a wealth of data and analyses (extensive phylogenies and genome analyses) to be explored in numerous ways by interested researchers who study symbiosis. It is a shame that some real gems are relegated to the supplementary files. The videos are wonderful. This paper will be of interest to the readers of Nature Communications and will be well-cited as a model study of symbiosis. The methodologies are sound, and work meets the expected standards in the field. Enough detail for methods is provided (see notes in comments below), and the work supports the conclusions. Some minor comments for the authors to consider:

We thank the reviewer for the supportive assessment of our work.

R1Q1. Abstract line 29: perhaps "highly specialized structures of symbiotic origin..."? To differentiate between the organelles of symbiotic origin and the current symbioses?

We agree. We have changed text to clarify what we meant: “highly specialized structures for housing symbionts”

R1Q2. Abstract line 47: “enhanced” instead of “enriched”?

Yes. We now use "enhanced".

R1Q3. Line 58 fix grammar

fixed

R1Q4. Line 72: Reference 4 is ok here, but this paper is mainly the genomics investigation Beinart made into this symbiosis, and the original report of the symbiosis containing more microscopy showing what you highlight in this sentence is the paper Edgcomb et al. 2011 Frontiers in Microbial Physiology.

Thank you for the suggestion. We have added this reference.

R1Q5. Line 85: consider making the definition of ‘symbiosome’ clearer at first introduction of the term since it is not frequently used yet. A reference or brief definition, or move line 91 up to help clarify.

We have now clearly defined the term symbiosome by adding the following sentences to lines 59-62, pg 3:

“If the symbionts are intracellular, such structures are often referred to as ‘symbiosomes’ that are defined as “membrane-bound compartment[s] housing one or more symbionts ... located in the cytoplasm of eukaryotic cells”³.

R1Q6. Line 101 and Figure 1B-E it would help to more clearly mark the vacuoles you are referring to for the reader in the figure. The arrows and boxes are very hard to see. Color maybe?

We have changed Figure 1 to better indicate the locations of the symbiosome by colouring Figure 1f and we have emphasized the boxes and arrows. In Figure 1d we added a marker for the food vacuole present in the *Anaeramoeba* cell.

R1Q7. Figure 2 legend for h-i: Please separate notations for each panel and check the color notations.

The color notations have been fixed.

It is confusing to have plasma membrane in red and then in yellow next sentence. I don't see red plasma membrane. Also clarify "Two symbionts whose membrane are contiguous with the plasma membrane are shown in red." Should this be blue?

Yes it should be blue. We have revamped Figure 2 h-i to show the two symbiosome subcompartments, altogether housing 108 symbionts, that are in contact with the outside media.

Last sentence of legend needs clarification too.

We have adjusted the sentence as follows: "Connections between symbiosome-membrane and plasma membrane are indicated by black arrows"

R1Q8. Figure 3. Suggestion that it would be nice to add negative control images for each probe in the supplementary since the area of the nucleus often shows signal for the bacterial probes. This can be due to orientation. The FISH protocol does not mention use of positive and negative controls nor Table S12.

We did include negative controls in our analyses. We provide an additional figure showing control experiments establishing the specificity of the BUSS/SCH probe. The new Figure S7 shows probing of *A. flamelloides* BUSSELTON2/SCHOONER1 using a reverse complemented probe (BUSS/SCH_RV), a scrambled probe (BUSS/SCH_SCR) of the same length and base composition as well as no probe control. Table S12 has been updated with information about the additional probes. For *A. ignava* BMAN we have lost the cultured isolate and are therefore unable to provide similar control experiments. However, we fail to see hybridization of the BMAN probe to the *Desulfobacter* sp. symbionts in *A. flamelloides* BUSSELTON2 and SCHOONER1.

R1Q9. Line 145: is it possible to say based on microscopy whether these estimates for numbers of symbionts per host are supported?

It is not straightforward to get a precise count of symbionts from the FISH analyses because the symbiont cells are tightly associated in large masses. Our FIB-SEM analyses indicate a larger number of symbionts per cell (185 for the particular cell showcased) than the genome sequence coverage average predicts. However, we see quite a large variation in symbiont numbers per cell since sizes in the culture vary (presumably because of different cells being in different cell cycle stages). Furthermore, the ploidy of the host nucleus is unknown. Different ploidies would affect the ratio of symbiont/host coverage. Additionally, the particular cell imaged for FIB-SEM may have been poised to undergo cell division. At this point, a precise investigation of this is beyond the scope of this paper (which is already extremely data-rich, as other reviewers have pointed out). But it is a topic we are actively investigating.

R1Q10. Line 218: You discuss the cob genes for B12 synthesis lower in the manuscript, present in the symbiont, and you say that ROS defense mechanisms exist in the symbionts, and so it would be also interesting if you looked for evidence of the anaerobic B12 synthesis pathway in the symbionts. The specific genes involved are *cbi*.

The genes we found for B12 are indeed encoding enzymes in the anaerobic pathway for B12 synthesis. We have modified the text to indicate the pathway we have found is the anaerobic one.

R1Q11. Lines 226-230: Can you comment here or in discussion on the significance of such high A+T?

The cause of the high A+T content of these *Anaeramoeba* genomes is unclear. It may be due to a strong bias in base substitution mutation types occurring as has been proposed for *Dictyostelium discoideum* that has a high A+T genome content ([Kucukyildirim et al. \(2020\) https://pubmed.ncbi.nlm.nih.gov/32732307/](https://pubmed.ncbi.nlm.nih.gov/32732307/)). However, large-scale mutation accumulation studies had to be done to make this conclusion. Without this kind of data for *Anaeramoeba*, there is little that can be definitively said about its A+T bias. Given space limitations, we would prefer to not dilute the focus of our paper with speculations on this topic.

R1Q12. Line 234: Section on B12 dependency. Perhaps point out that in some cases, B12 can also provide methyl groups (methylcorinoids) that support C1 metabolism? See Author summary in Price et al. 2021.

We thank the reviewer for this interesting point. Price *et al.* 2021 describes that B12 derivatives can provide a methyl group in the cysteine synthesis reaction when *MesA*, *MesB* or *MesC* are the methionine synthases. However, we did not find any of the genes encoding these enzymes in the *Anaeramoeba* spp. genomes. While it is still possible that B12 derivatives are providing chemical groups in other metabolic processes in *Anaeramoebae*, because we have no clear evidence for it, we did not add this possibility to the text.

R1Q13. Line 347 on: where you describe Rabs and their function, please acknowledge their possible role in signal transduction and as molecular switches that regulate formation and transportation of vesicles (e.g., Hutagalung and Novick 2011).

We have modified the line in question to acknowledge these points and included the reference.

R1Q14. Any evidence for archaeal symbionts in Metamonada? Thinking of your LGT data.

This is a good question. Within the Metamonada, methanogenic archaea are known as ectosymbionts of termite-associated parabasalids and oxymonads where they act as hydrogen scavengers associated with host cytoplasmic membrane-associated hydrogenosomes.

To address this we investigated our LGT phylogenies more closely and found that there was no particular 'signal' for genes acquired specifically from methanogenic archaea. Because the manuscript is already very long and detailed, we have opted not to elaborate on the archaeal transfers in the revision.

R1Q15. Lines 408-410: They maintain connections to the outside environment presumably due to metabolic constraints of the symbionts that require sulfate, however is it possible also that this is also maintained by modifications/restrictions in the membrane-trafficking system of the host that prevent the vacuoles from fully forming and hence cutting them off from direct contact with the outside environment? Your results section suggests this might be the case, so perhaps mention here? I apologize if in the extensive documents I missed this!

We have expanded our discussion on this point as follows (lines 434-440, pg. 17):

"In the Anaeramoebae, it remains unclear the degree to which the cell surface connections to the symbiosomes are relatively static or continuously forming and closing. Regardless, given the many fine connections between individual symbiont-housing subcompartments (Figure S4) there appears to be continuity in the lumen of a large majority of these subcompartments (Figure 2 H-I, Figure S4-5) allowing exchange of metabolites between symbionts and with the external environment. The WGA staining experiment indicates that this exchange is rapid and extensive."

R1Q16. Lines 514-7: Please clarify for the reader why different assemblers and min. overlaps were used for the three host genomes.

The genome sequencing was completed in multiple steps at different times. The rapidly changing software landscape for assembling long-read and short-read data means that different assemblers were available (and better optimized for the data) at these different times. Some assemblers, such as Unicycler, are designed for use with bacterial genomes only. This is why such a diversity of assemblers were used in the project.

The overlap parameters were automatically chosen by the various assemblers based on the average read length and depth of coverage.

R1Q17. Table 1. Can you perhaps clarify what exactly the column labeled "trees" is showing either by changing the column header or in the legend?

We have now indicated in the footnote to the table that the Trees column indicates the trees are estimated from a subset of OGs that contained at least one prokaryotic or viral taxa and met the criterion of having >80 sites.

R1Q18. I may have missed it, but either in the reporting summary or text documents, please be sure that readers know how /where to acquire cultures of these organisms if desired.

It is stated in the methods section under the subheading *Culturing and harvesting of Anaeramoeba* that “Cultures of *Anaeramoeba* strains are available upon request.”

R1Q19. Any reason why Delta495a probe images are not available in Fig. S6 for *A. flamelloides* BUSSELTON2 or SCHOONER?

We have added the Delta495a probe images from *A. flamelloides* BUSSELTON2 and SCHOONER into Fig. S6 (now Figure S8).

R1Q20. Supplementary discussion of selenoproteins: Perhaps point out that use of selenoproteins and selenocysteins while present in certain anaerobic taxa is overall not a common trait, and their synthesis may be a trait acquired through horizontal transfer events (Peng et al. 2016).

This is a good point. We have added the following text to lines 261-262, pg. 11 of the Supplementary text:

“The ability to make selenocysteine and selenoproteins is not a universal feature and it may be acquired by horizontal transfer events (Peng et al. 2016).”

Reviewer #2 (Remarks to the Author):

In this manuscript by Jerlström-Hultqvist and colleagues, the authors examined the symbiosis between multiple *Anaeramoeba* species and their *Desulfobacteraceae* symbionts. The manuscript is a methodological tour de force using most methods currently available for studying symbioses of protists. The authors generated not only complete genomes of several symbionts but also transcriptomes, high-quality host genomes, and multiple types of imaging data. The bioinformatics analyses are solid, and address a broad range of questions, including LGT and gene family expansions in the nuclear genomes. I do not find it necessary to comment on the genomics part of the manuscript because it's already very polished and on par with the highest standards in the field. Overall, my evaluation of the manuscript is very positive. My only suggestions for improvement concern the microscopy results, the appropriateness of the term symbiosome, and the manuscript not being accessible to a broad audience.

R2Q1. I find the manuscript overwhelming and not very accessible to a broad audience. The authors did a great job of running all possible analyses. However, they included all the results in the MS which means that there's so much data that many interesting novel findings are buried in supplementary files (15 supplementary tables and 19 supplementary figures). The amount of data included in this paper, in my view, goes over the limits of what should be included in a

single research article (it reads more like a monograph on everything we know about this particular symbiosis) which inevitably means that some results are only touched superficially, and a broad synthesis is missing.

We appreciate the reviewer's concerns. Our goal was to comprehensively address the nature of this symbiosis in a single manuscript rather than publishing smaller parts of this work piecemeal. While we agree that, in the end, this means a large amount of different data and analyses are included here, we have tried very hard to highlight in the main text the most important findings from our comprehensive analyses. At this stage, we believe that it is not feasible, nor desirable, to break this work up into multiple different manuscripts. Furthermore, we believe that having a single manuscript reporting these data will facilitate timely release of the data.

R2Q2. The vacuole/organelle housing all the symbionts is presented as the main finding of the manuscript. Like the term 'bacteriome' that is used broadly in multicellular eukaryotes for diverse organs that are composed of cells (bacteriocytes) housing endosymbionts, the term symbiosome is used here for a novel membranous organelle harboring symbionts. I fully agree that this finding is potentially extremely interesting. However, extraordinary claims require extraordinary evidence. The result is only supported by limited imaging data and is not properly discussed in the context of what exactly the organelle is. First, I am not 100% convinced that the single FIB-SEM dataset provides sufficient evidence that a single vacuole houses all the symbionts.

To address this concern, we further analyzed the FIB-SEM data manually by checking each symbiont subcompartment in all dimensions to trace connections between them. In total 171 connections between subcompartments were identified (see heavily expanded Figure S4). We also included a new animation depicting these connections (see new Animation S2). We found 15 sets of connected subcompartments, i.e., subcompartments with tubular connections to other subcompartments. The largest set of subcompartments included 105 of the symbionts in a large connected network (see Figure 2 H-I and new Figure S5). This subcompartment of 105 symbionts and a second subcompartment of 3 symbionts are connected to the cells exterior as displayed in Figure 2H-I. The other 14 connected sets included 2-10 symbionts. There were an additional 25 symbiont subcompartments that lacked a traceable connection to another subcompartment. Animation 2 shows the different subcompartments and labels for each connection. Even though we were unable to find contacts between all the symbiont subcompartments, 85% of symbionts are connected to at least one other symbiont via tubular connections. There might be several contributing reasons why we did not find a fully connected symbiont network.

- i) Given the exceptionally small dimensions of the inter-subcompartment connections, which often occur in single or a few FIB-SEM slices, there might be additional contacts that eluded detection.
- ii) The symbiosome subcompartments are likely dynamically connecting and disconnecting with each other. This is certainly the case during cell division when

the subcompartments are divided and segregated into daughter cells (see Figure S1).

- iii) We see many examples of 'futile' membrane extensions from symbiosome cell compartments that ultimately do not reach a neighboring subcompartment. These could represent precursors to, or remnants of, connections.

To account for these new data and interpretation, we have added an expanded treatment of the methods and results in the main text lines 119 to 140, lines 727-754 and in the "FIB-SEM extended results" section of the Supplementary text.

R2Q3. Could you please expand the methods section with details on how exactly were the data segmented and what is the pixel size?

A comprehensive description of the segmentation and other information relating to the FIB-SEM analysis is given in the Supplementary text, line 29 under "FIB-SEM extended methods and results". We have added additional information regarding this in the Materials and Methods in the main text as well in a new section "Segmentation and visualization of cell structures". The pixel sizes are equal to the section thickness and determine the resolution. Our pixel sizes are almost isotropic at $(6.7 \times 6.7 \times 7 \text{ nm})$. This means that the smallest volumetric unit (voxel) is $6.7 \times 6.7 \times 7 \text{ nm}^3$.

R2Q4. If the resolution was sufficient for confidently segmenting membranes, why is the endomembrane system not shown?

Segmenting the entire endomembrane system is an exceptionally difficult task because of the extensive networks of fine tubules and the lack of resolution at that scale. It was difficult enough to segment the symbiosome-membrane network and train the machine learning methods to reliably recognize its features and, even then, further optimization by extensive manual correction was necessary.

R2Q5. Does the 'symbiosome' somehow interact with (or is a part of) the ER and Golgi? '

This is currently unknown and is the subject of ongoing follow-up studies. It should be noted that *Anaeramoeba* does not have a recognizable Golgi or dictyosomes (see reference 6, Táborský et al. 2017). As was pointed out above (R2Q1), there is already an exceptionally large amount of data in this study and we think it is unwise to expand the scope of our analyses further.

R2Q6. Are any other compartments (such as food vacuoles) of *Anaramoeba* also partly connected to the extracellular environment?

We do observe food vacuoles in both the traditional EM and in the FIB-SEM data. In fact what appears to be a large food vacuole/phagosome with a connection to the surface is shown in Figure 1d (labelled 'fv'). The symbiosome appeared to be quite separate from these vacuoles and no connections between symbiosome or putative food vacuoles were observed during

manual tracing of symbiont subcompartment contacts. Two degenerating symbionts can be observed in the volume.

R2Q7. What happens with the symbiosome after symbiont loss/replacement?

This is currently unknown and is the subject of ongoing follow-up studies.

R2Q8. Expanding the symbiosome definition to apply to the *Anaeramoeba* system is, in my view, potentially misleading. The term symbiosomal membrane is used in many different systems (including protists) for a membrane of host origin surrounding a single symbiont cell (and symbiosome is used, relatively rarely, for the entire compartment). Please correct me if I'm wrong, but this seems to be the first time the term symbiosome is used for a host-derived vacuole network surrounding many symbiont cells.

The term "symbiosome" was originally defined in "Homology in endosymbiotic systems: the term 'symbiosome'" by Roth, Jeon and Stacey (1988) (now cited as reference #3) as:

*"A membrane-bound compartment containing **one or more symbionts** and certain metabolic components and located in the cytoplasm of eukaryotic cells. "* [our emphasis]

This definition indicates that symbiosomes can house multiple symbionts. In fact this paper discusses a few protist-bacterial symbioses as examples where there are multiple symbionts within host 'symbiosomes'. For example, Roth Jeon and Stacy showcased the seminal work on the X-symbiont in the D-strain of *Amoeba proteus* by Jeon and colleagues where multiple symbionts are housed within symbiosomes (see Fig. 1, Jeon et al. (1987); DOI: [10.1111/j.1749-6632.1987.tb40622.x](https://doi.org/10.1111/j.1749-6632.1987.tb40622.x)).

In our view, it is appropriate to use the symbiosome for the symbiont-housing compartment in *Anaeramoeba* since the symbionts are almost entirely enclosed within the cytoplasm of the cell. For ease of communication we would prefer to use "symbiosome" rather than invent a completely novel term to describe the structure.

We note that the Roth, Jeon and Stacy paper also introduces the term "symbiosome-membrane" for the host-derived membrane and "symbiosome space" for the space enclosed by the symbiosome-membrane that is not occupied by symbionts. For consistency with these original definitions, we now use these latter terms throughout the manuscript.

R2Q9. Since it has tubular connections leading to the cell surface, its origin and membrane composition are, however, less clear. Did it originate via a similar process that leads to what we usually call a symbiosomal membrane?

Given that symbiosomes have separately evolved in plants, various metazoan lineages and protist lineages, it is very likely that each of these independently evolved structures originate by different processes (or differ in specific details in how they originate). In any case, the processes by which symbiosomes originate in most taxa is simply unknown. Addressing this question for *Anaeramoebae* is the subject of ongoing research in our groups.

R2Q10. The authors suggest that the organelle is reminiscent of phagosomes that have been delayed or frozen in their maturation process (P14L347) but no experimental evidence supporting this hypothesis is included.

Here we propose a hypothesis that can be tested in the future. We do not claim to have experimentally demonstrated this.

R2Q11. In well-studied systems such as the Buchnera-pea aphid model, the symbiosomal membrane originates when extracellular symbionts are endocytosed by the posterior syncytial cytoplasm of blastula (Koga et al., PNAS 2012). Buchnera cells then divide into daughter cells that are both covered by the 3rd symbiosomal membrane. In the case of Anaramoeba, the symbiont cell division happens inside the vacuole which could suggest that this membranous organelle is different from what would be usually called a symbiosome. The authors should clarify what the organelle is and use the term consistently.

We have defined our use of the term “symbiosome” in the context of *Anaeramoeba* by reference to its original definition as detailed in our response to question (R2Q8). We now apply this term and associated terms defined by Roth, Jeon and Stacey (1988) consistently throughout the manuscript.

R2Q12. Some of the descriptions from the text (see below) are too vague or even contradictory (vacuoles, P5L99).

We have changed the listed descriptions of the symbiosomal network to be more consistent.

For example, on P5L99 (lines 104-106 on pg. 5 in the revision) we have changed the sentence to avoid using the term vacuole:

“Near the nucleus in the bulbous cell body lies a large mass of symbionts apparently housed in compartments with single bounding membranes tightly positioned close to, but not in direct contact with, hydrogenosomes”

P2L36: We show that the symbionts reside deep within a symbiosomal membrane network that is tightly associated with cytoplasmic hydrogenosomes, and, importantly, maintains connections to the Anaramoeba cell surface.

“cell surface” changed to “plasma membrane”.

P4L79: These symbionts, although not in direct contact with the hydrogenosomes, are enveloped by an intricately organized membrane of host origin.

“intricately organized” removed.

P4L91: ...the symbionts are situated within a symbiosome, which comprises an interconnected vacuolar network positioning the symbionts alongside host hydrogenosomes.

“vacuolar” changed to “membrane”.

P5L99: ... a large mass of symbionts apparently housed in vacuoles tightly positioned close to, but not in direct contact with, hydrogenosomes ...

“symbionts apparently housed in vacuoles tightly positioned”

changed to

“symbionts housed in compartments with single bounding membranes tightly positioned”

P6L128: ... a network of deep, anastomosing invaginations of the cell’s membrane with extracellular opening.

We have removed this sentence as it was unnecessary.

P16L397: ...suggests that the membranous ‘organ’ housing the symbionts is a highly evolved structure that might allow them to selectively manage their captured symbionts and position them in tight association with their hydrogenosomes.

We updated the sentence on line 554-555 of the revised MS.

“suggests that the membranous ‘organ’ housing the symbionts is a highly evolved”

changed to

“suggests that the membrane compartment housing the symbionts is a highly evolved”

Reviewer #3 (Remarks to the Author):

This is an exciting and broadly interesting paper. I agree with the authors that not much is known about the cell biological structures of symbiotic systems, protists in particular, and that this paper provides a unique look into a very interesting protist-bacterial symbiosis. It is novel but also provides interesting comparisons with (comparatively better-studied) animal-microbe and plant-microbe symbioses. The paper has already changed the way I view symbiosis, by giving me a cool example of the cell biological Rube Goldberg machines host can make to enable their symbioses to work. This paper will be widely read, discussed, cited. It’s exciting and high-quality work.

That being said, I do have one major issue, one minor (stylistic) issue, and several small points. I believe all of these points are addressable without further experiments, but might require

making one or two more figures, as I explain below.

R3Q1. Major issue:

The critically important result that the bacteria exist in a connected network that has continuous exposure to the outside world is not clearly shown in the paper. The WGA result is very suggestive and I think a good experiment. The TEM in Figure 1f does show very close connections between hydrogenosomes and the host vacuolar membrane, but in this case the “narrow opening connected to outside media” could be a phagocytic event that is not yet complete. It *could* be a completed phagocytic event in which a connection to the outside world is retained, but it also might not be. One can see other bacteria that seem to be undergoing phagocytosis nearby (the bacterium at the lower-right corner of the dashed inset box, for example) and so in my mind this complicates this interpretation.

The 3D reconstructions of the FIB-SEM data in Figure 2 do not show this, either. Figure 2h does show connections to the outside world (the black arrows), and I agree that the shape of these connections are less likely to be captured in-process phagocytic events, but nowhere in Figure 2 are the networks of connections shown.

The closest the paper gets to showing the network is in Figure S4. Here, pairs of clearly interior bacteria are seen to have openings which might connect these bacteria to each other. But nowhere (that I can see) are connections traced from the surface to bacteria that are well inside the protist.

I strongly suspect that these data exist, and that the authors have seen these connections. (I was hoping to see this in the videos, but I didn't there, either.) The quality of the FIB-SEM and the TEM is good enough that I am certain this could be shown in the main paper. I would suggest that Figure 2 be re-worked. Some existing panels could be put into supplemental (maybe d,e,g, and i?) which would make space for a reconstruction that traces the plasma membrane, through a surface-adjacent symbiosome, to a neighboring symbiosome, to another, etc., such that the host-membrane connectivity between outside and interior symbionts is very clear.

As mentioned in response to reviewer #2 we have added a comprehensive analysis of symbiont subcompartment connections. We have reworked Figure 2 panels H and I to show the symbiosome subcompartments that are connected to the plasma membrane. We added a heavily expanded Figure S4 that shows 171 connections we observed between the symbiosome subcompartments. The cellular context of those 171 subcompartment connections and their connectivity are shown in a new Animation 2. The subcompartments are visualized in 3D in a new Figure S5.

As the paper is now, a reader can somewhat piece this together, but the conclusion of complete interconnection now rests mostly on trust of the authors (of which I have a great deal, to be honest).

As discussed in response to reviewer #2, we have obtained substantially more evidence for sets of interconnected subcompartments in the symbiosome and present these data in a number of new figures and discussion. The subpanels H and I of Figure 2 explicitly depict the groups of symbionts that are in sets of interconnected subcompartments with connections to the outside of the cell. All of the individually catalogued connections are now shown in Figure S4 and Animation 2 depicts subcompartment connections as you move through the segmented FIB-SEM volume.

R2Q2. Minor issue:

The paper starts off strongly, with (mostly) experimentally and visually compelling data and results. The following sections on genomics are a bit of a tougher read, and seem about 50% too long. There is interesting stuff in there, and I think the main results should stay in the paper because they obviously contextualize the imaging, but many of the details could be moved to supplemental. Alternatively, maybe the genomics could be presented first, in a shorter format, to set up the question of how the symbiosis could possibly work when the symbionts need both tight connections to hydrogenosomes AND connection to the outside world? People like riddles.

I can see it either way, and it's not my paper, but that's my honest assessment.

We appreciate the reviewer's comments but have chosen not to change the order of our manuscript. However, in accordance with the suggestion we have streamlined the various sections that the reviewer suggested are too long.

Small points:

R2Q3. 1. The abstract mentions that the data supports the idea that the host "facilitates symbiont partitioning during cell division." I am not sure the data strongly support this point being included in the abstract, but perhaps I missed something.

The data to which the sentence in question is referring is in Supplementary Figure 1. However we realize the evidence to support this claim is limited so we have removed this phrase from the abstract.

R2Q4. 2. What is the interpretation of Figure 1b? What are we to learn from it?

Figure 1b is included to depict the general organization of the cell and its microtubular cytoskeleton. We have included a sentence in the results to better describe this figure panel.

“*Anaeramoeba* cells have an acentriolar centrosome positioned below the hydrogenosome-symbiont mass and the nucleus from which microtubules radiate throughout the ventral side of the cell (Figure 1B, E).”

R2Q4. 3. Figure legend for Fig 2h-I refers to red color but I think this is a mistake.

Yes. We have fixed this now.

R2Q5. 4. I had to look up anastomosing, which is fine, and I’ve learned something, but is there a simpler way to say this?

We would prefer to use the term anastomosing because it conveys information that otherwise requires multiple words to communicate.

R2Q6. 5. Lines 441-44: “If *Anaeramoeba* species are truly dependent on symbiont-provided vitB12, then this might also mean that cyclical symbiont-replacement, as seen in the ciliate *Euplotes* 41, is necessary to ensure that they receive a steady source of vitB12 in the face of symbiont genome erosion.” Maybe, but it seems like if B12 is important genes for it would be under at least as strong purifying selection as all other genes important to the host, and so I don’t know why you would expect turnover based on these genes in particular.

We agree and have deleted this sentence.

R2Q7. 6.Line 445: Yes, this does support the shopping bag idea, but so also does the results from both other protist (10.1016/j.cub.2017.08.010) and insect (10.1016/j.cell.2013.05.040) symbioses. Would be good to contextualize these results a bit more with the literature.

We have added a bit more context to the sentence:

“Our LGT analyses indicate that no single bacterial lineage has disproportionately contributed to the gene repertoire of *Anaeramoeba*, which is reminiscent of other symbiotic systems in protists (Singer et al. *Curr Biol.* 2017 Sep 25;27(18):2763-2773.e5..) and insects (Husnik et al. *Cell.* 2013 Jun 20;153(7):1567-78). Our data and these other examples are consistent with the shopping bag model of gene acquisitions in relation to the evolution of symbiont-derived organelles ⁴²”